# Any-Dimensional Invariant Universality

**Shengtai Yao** [1]  **Eitan Levin** [2]  **Mateo Díaz** [2 3]

## Abstract

Several machine learning models are defined for inputs of any size, such as graphs with different numbers of nodes and point clouds containing varying numbers of points. The universality properties of such any-dimensional models remain poorly understood, as universality is traditionally studied for models accepting inputs of a fixed size, defined on a compact subset of their domain. In sharp contrast, any-dimensional models can be viewed as sequences of functions defined on growing-sized inputs, and it is not clear in which sense they can be universal. We develop a systematic approach to establish any-dimensional universality, by identifying any-dimensional functions with a unique function taking inputs in a suitable infinite-dimensional limit space containing inputs of all finite sizes as well as their limits. Using the symmetries of these inputs and relations between inputs of different sizes, we show that this limit space admits a natural topology with rich families of compact sets on which any-dimensional universality can be established. We illustrate our approach by showing that several existing architectures fail to be universal, and we propose simple modifications that restore universality.

## 1. Introduction

Traditional supervised learning aims to learn mappings defined on fixed, finite-dimensional spaces by using training examples lying in those same spaces. In contrast, many modern learning tasks involve maps defined on inputs of varying dimensions. For instance, graph parameters, particle system dynamics, games, and regularizers remain meaningful regardless of the number of nodes, particles, players, or variables. There are a number of architectures in

the literature that are defined on inputs of any dimension, e.g., DeepSets (Zaheer et al., 2017) for sets, Graph Neural Networks (GNN) (Gori et al., 2005) for graphs, and Point-Net (Qi et al., 2017) for point clouds. Effectively, these any-dimensional architectures finitely parametrize an infinite sequence of invariant functions $(\widehat{f}_n \colon V_n \to \mathbb{R})_n$ with increasingly larger input dimensions.[1] Despite the proliferation of any-dimensional architectures, their expressivity remains relatively unexplored. Specifically, the universality of most of the any-dimensional architectures above has mostly been studied in a fixed dimension or in finitely many dimensions (Maron et al., 2019b; Keriven & Peyré, 2019; Yarotsky, 2022). This motivates our main question

> *Can any-dimensional models universally approximate functions across dimensions?*

As stated, this question is ill-posed. To understand why, recall that classical universality concerns approximating a *single* continuous function $f \colon K \to \mathbb{R}$ on a fixed compact set $K$ by a model $\widehat{f}$ uniformly on $K$. Any-dimensional models, however, produce a *sequence* of maps with varying domains. To resolve this mismatch, we adopt the framework of (Levin et al., 2025). Specifically, we view the input spaces as a nested sequence $V_1 \subseteq V_2 \subseteq \cdots$ of increasing-dimensional vector spaces, and embed them all into an infinite-dimensional limit space $V_\infty$ we construct containing each $V_n$ as a subspace. Under a mild compatibility condition, an any-dimensional model $(\widehat{f}_n)$ admits a unique extension $\widehat{f}_\infty \colon V_\infty \to \mathbb{R}$ satisfying $\widehat{f}_\infty(x) = \widehat{f}_n(x)$ for any $x \in V_n$ and $n$. This identification lets us pose "universality across dimensions" as standard universality on an application-specific infinite-dimensional space.

**Main contributions.** Let us summarize our contributions.

(***A recipe***) We develop a general strategy for proving any-dimensional invariant universality. The strategy highlights two principles: $(i)$ exploiting symmetries by passing to orbit spaces can yield rich families of compact sets, even in infinite dimensions, on which universality can be proved;

---

[1]Department of Management Science and Engineering, Stanford University. [2]Department of Statistics, [3]Data Science Institute, University of Chicago. Correspondence to: Mateo Díaz <mateodd@uchicago.edu>.

*Proceedings of the 43rd International Conference on Machine Learning*, Seoul, South Korea. PMLR 306, 2026. Copyright 2026 by the author(s).

---

[1]By 'invariant' we mean that there is a sequence of groups $(G_n)$, such as permutations or rotations, such that for each $n$ the group $G_n$ acts on $V_n$ and for all $g \in G_n$ and $v \in V_n$, we have $f_n(g \cdot v) = f_n(v)$.

and $(ii)$ unlike in finite dimensions—where all norms induce the same topology—the choice of norm on $V_\infty$ is essential, as continuity (and hence admissible activations and architectural primitives) depends on it.

(***Instantiations***) We apply this recipe to three representative domains: sets, graphs, and point clouds. We show that several widely used architectures yield extensions $\widehat{f}_\infty$ that are either discontinuous in the natural topology induced by the learning task or fail to be universal. We then propose modifications—and, when necessary, new architectures—that restore continuity and achieve universality.

**Outline.** The remainder of this section reviews related work. Section 2 reviews the mathematical preliminaries of any dimensional learning necessary for the rest of the paper. Section 3 formalizes the type of any-dimensional universality that we wish to establish and outlines a general recipe for establishing universality. Section 4 presents concrete instantiations of this recipe for models that apply to sets, graphs, and point clouds. Finally, Section 5 concludes the paper with limitations and opportunities for future work.

### 1.1. Related work

**Dimension-free learning.** Many modern architectures are any-dimensional. Convolutional neural networks apply the same translation-invariant filters to images of arbitrary resolution (LeCun et al., 2002; Hashemi, 2019). Recurrent neural networks and transformers handle variable-length sequences via recurrence or attention (Hochreiter & Schmidhuber, 1997; Socher et al., 2011; Vaswani et al., 2017). Neural operators such as DeepONet and FNO learn maps between (infinite-dimensional) function spaces and can be evaluated on grids of different resolutions (Lu et al., 2021; Li et al., 2020). In geometric deep learning, several models have been proposed to handle sets of arbitrary cardinality (Zaheer et al., 2017; Qi et al., 2017), graphs with varying numbers of nodes and edges (Gori et al., 2005; Scarselli et al., 2008), and point clouds with any size (Villar et al., 2021; Blum-Smith et al., 2025). Recently, (Levin & Chandrasekaran, 2023; Levin & Díaz, 2024; Díaz et al., 2025) leveraged *representation stability* (Church & Farb, 2013; Church et al., 2015) to derive general-purpose dimension-free equivariant and invariant model classes, encoding neural networks, convex sets, and kernel machines. However, the ability to accept variable sizes does not guarantee consistent (or transferable) behavior across sizes. The study of transferability originated in the GNN literature (Ruiz et al., 2020) and has since been explored extensively in that context (Levie et al., 2021; Keriven et al., 2020; Ruiz et al., 2023; Maskey et al., 2023; Cordonnier et al., 2023; Cai & Wang, 2022). More recently, (Levin et al., 2025) proposed a unified framework—beyond graphs—for formulating and certifying transferability across dimensions.

**Universality.** Universality is one of the pillars of deep learning theory. Let us start by commenting on finite-dimensional results. Classical universal approximation theorems show that fully connected networks approximate continuous functions on compact sets (Cybenko, 1989; Hornik et al., 1989; Leshno et al., 1993; Hornik, 1991), with recent extensions beyond compact domains under suitable growth and activation assumptions (van Nuland, 2024; Neufeld & Schmocker, 2024; Abdeljawad & Dittrich, 2024). The expressivity of symmetric models has only recently been studied. For graphs, expressive power is subtle: message-passing GNNs are known to have intrinsic limitations (Xu et al., 2018), commonly formalized via the Weisfeiler–Lehman (WL) test (Leman & Weisfeiler, 1968), and many variants have been analyzed through this lens (Xu et al., 2018; Maron et al., 2019a; Morris et al., 2019). Alternative GNN based on tensor layers (Maron et al., 2019b; Keriven & Peyré, 2019), and those based on homomorphism density functions (Maehara & NT, 2019; Nguyen & Maehara, 2020) do achieve universality. For permutation-invariant problems, DeepSets is universal (Zaheer et al., 2017; Sannai et al., 2019). Beyond these specific examples, (Yarotsky, 2022) established universality of polynomial-invariant-based models that are symmetric under the action of any compact group. Nonetheless, these results are largely formulated for a fixed input dimension. By contrast, universality for any-dimensional models remains comparatively underdeveloped. The closest related results are (Bueno & Hylton, 2021; Zweig & Bruna, 2021) for sets and (Keriven & Peyré, 2019; Herbst & Jegelka, 2025) for graphs. Our results and strategy highlight a more general perspective based on the transferability framework developed in (Levin et al., 2025), applicable beyond these concrete scenarios. Let us briefly comment on more specific differences. First, (Zweig & Bruna, 2021) studies lower bounds on approximation for certain RKHS classes; these classes are not our focus. Our DeepSets results extend (Bueno & Hylton, 2021) by covering a broader class of compact sets. On the graph side, (Keriven et al., 2020) proves universality of graphon neural networks in simplified regimes, while (Herbst & Jegelka, 2025) establishes universality with respect to the $\delta_p$ metric. In contrast, we work with the arguably more natural and widely studied cut metric $\delta_\square$, which induces a coarser topology.

## 2. Preliminaries

In this section, we introduce the notation and technical background necessary for the paper.

**Notation.** We use $\mathbb{R}$ and $\mathbb{N}$ to denote the reals and positive integers, respectively. Define $\mathbb{N}_0 = \mathbb{N} \cup \{0\}$ and $[n] := \{1, \ldots, n\}$ for $n \in \mathbb{N}$. We write $\mathrm{O}(k)$ for the orthogonal group in dimension $k$ and $\mathrm{S}_n$ for the symmetric group on $n$ letters. For a given metric space $(\mathcal{X}, d)$,

we denote the open ball centered at $x$ of radius $\eta$ by $B(x, \eta) := \{u \in \mathcal{X} \mid d(u, x) < \eta\}$. Let $C(\mathcal{X}, \mathcal{Y})$ denote the space of continuous maps from a topological space $\mathcal{X}$ to $\mathcal{Y}$, and write $C(\mathcal{X})$ when $\mathcal{Y} = \mathbb{R}$. Let $\mathcal{B}(\mathcal{X}, \mathcal{Y})$ be the space of bounded linear operators between normed spaces $\mathcal{X}$ and $\mathcal{Y}$. Given an arbitrary norm $\|\cdot\|_{\mathbb{R}^k}$ on $\mathbb{R}^k$, the symbol $\ell_p(\mathbb{R}^k)$ denotes the space of sequences $X = (X_{j:} \in \mathbb{R}^k)_{j \in \mathbb{N}}$ such that $\sum_{j=1}^{\infty} \|X_{j:}\|_{\mathbb{R}^k}^p < \infty$. Similarly, for a given measure space $(\mathbb{R}^m, \mu)$, let $L^p(\mathbb{R}^m; \mathbb{R}^k)$ be the space of measurable functions $X : \mathbb{R}^m \to \mathbb{R}^k$ with $\|X\|_p := \left(\int \|X(t)\|_{\mathbb{R}^k}^p d\mu(t)\right)^{1/p} < \infty$. The symbol $\|\cdot\|_p$ refers to either the $\ell^p$- or $L^p$-norms, as appropriate. Finally, let $\mathcal{P}(\mathbb{R}^k)$ be the set of probability measures on $\mathbb{R}^k$, and let $\mathcal{P}_p(\mathbb{R}^k) \subseteq \mathcal{P}(\mathbb{R}^k)$ denote those with finite $p$th moments. We use the symbol $W_p$ to denote the Wasserstein-$p$ distance.

**Any-dimensional learning.** Next, we recall a few basic notions pertaining to any-dimensional models and their generalization across dimensions. Our exposition here follows (Levin et al., 2025); we refer the interested reader to that reference for additional details. We focus exclusively on real-valued functions, and accordingly streamline several definitions, though all notions in this section extend to more general codomains. The key idea of this section is that a sequence of functions $(f_n : V_n \to \mathbb{R})$ with growing-sized inputs can be identified with a single extension $f_\infty : \overline{V_\infty} \to \mathbb{R}$, defined on an infinite-dimensional space $\overline{V_\infty}$ that 'contains' each $V_n$ as a subspace. This viewpoint lets us pose questions such as universality on one common domain, rather than across a sequence of domains. To state these ideas precisely, we start from a family of vector spaces encoding inputs of growing size, together with maps and relations linking them.

**Definition 2.1.** *A **consistent sequence** is a triple $\mathbb{V} = \{(V_n)_{n \in \mathbb{N}}, (\varphi_{N,n})_{n \preceq N}, (\mathrm{G}_n)_{n \in \mathbb{N}}\}$ indexed by a directed poset $(\mathbb{N}, \preceq)$,[2] of finite dimensional vector spaces $V_n$, maps $\varphi_{N,n}$ and groups $\mathrm{G}_n$ acting linearly on $V_n$ such that, for all $n \preceq N$, $(i)$ the group $\mathrm{G}_n$ is embedded into $\mathrm{G}_N$ and $(ii)$ $\varphi_{N,n} : V_n \hookrightarrow V_N$ is a linear, $\mathrm{G}_n$-equivariant embedding.*

The elements of different vector spaces in a consistent sequence represent objects of different sizes, while the groups and embeddings between them represent their symmetries and relations between objects of different sizes, respectively. To crystallize this notion, we provide two examples of consistent sequences that will play a crucial role in this work.

**Example 2.2.** *Let $V_n = \mathbb{R}^n$ and $\mathrm{G}_n = \mathrm{S}_n$ be the symmetric group acting by coordinate permutation. One might endow this sequence with two different types of orderings and embeddings.*

*1. (**Zero-padding embedding**) Index by $\mathbb{N}$ with the stan-*

---

*dard ordering $\leq$. Define $\varphi_{N,n} : V_n \hookrightarrow V_N$ by*

$$\varphi_{N,n}(x_1, \ldots, x_n) := (x_1, \ldots, x_n, \underbrace{0, \ldots, 0}_{(N-n) \text{ zeros}}).$$

*We use the symbol $\mathbb{V}_{\mathrm{zero}}$ to denote this sequence.*

*2. (**Duplication embedding**) Index by $\mathbb{N}$ with $n \preceq N$ if $n$ divides $N$. Define $\varphi_{N,n} : V_n \hookrightarrow V_N$ by repeating each coordinate $N/n$ times,*

$$\varphi_{N,n}(x_1, \ldots, x_n) = (\underbrace{x_1, \ldots, x_1}_{N/n \text{ copies}}, \ldots, \underbrace{x_n, \ldots, x_n}_{N/n \text{ copies}}).$$

*Similarly, we use $\mathbb{V}_{\mathrm{dup}}$ to denote this sequence.*

To define and study continuous any-dimensional models, we further define a single, infinite-dimensional space containing inputs of all finite sizes. This is done in the following definition.

**Definition 2.3.** *Define $V_\infty$ as the disjoint union $\bigsqcup V_n$ modulo an equivalence relation*

$$V_\infty := \bigsqcup_n V_n / \sim,$$

*where $v \sim \varphi_{N,n}(v)$ whenever $n \preceq N$, and denote by $[v] \in V_\infty$ the equivalence class of $v \in V_n$. The limiting group $G_\infty$ and its equivalence classes $[g]$ are defined analogously.*

In turn, this allows us to identify any finite-dimensional element $v \in V_n$ with its infinite-dimensional counterpart $[v] \in V_\infty$. The next definition will allow us to extend a sequence of functions to this infinite dimensional space.

**Definition 2.4.** *Let $\mathbb{V} = \{(V_n), (\varphi_{N,n}), (\mathrm{G}_n)\}$ be a consistent sequence indexed by $(\mathbb{N}, \preceq)$. A sequence $(f_n : V_n \to \mathbb{R})$ is **compatible** with respect to $\mathbb{V}$ if $f_N \circ \varphi_{N,n} = f_n$ for all $n \preceq N$, and each $f_n$ is $\mathrm{G}_n$-equivariant.*

In other words, a compatible sequence of functions is one that takes the same value on "equivalent" inputs, i.e., those related by symmetries and embeddings. A sequence of functions $(f_n)$ is compatible if and only if there exists a unique $G_\infty$-equivariant map $f_\infty : V_\infty \to \mathbb{R}$ such that $f_\infty|_{V_n} = f_n$ for all $n \in \mathbb{N}$ (Levin et al., 2025). Thus, we can identify any compatible sequence of functions with a unique extension that matches the functions when restricted to finite dimensions.

In order to reason about continuity of $f_\infty$, we will endow $V_\infty$ with a metric. To do so, we consider a sequence of compatible invariant norms $(\|\cdot\|_{V_n} : V_n \to \mathbb{R})$. Compatibility yields the existence of a $G_\infty$-invariant norm $\|\cdot\|_{V_\infty}$ on $V_\infty$. This construction allows us to define a limit space that contains not only (equivalent classes of) finite-dimensional objects, but also their limits.

**Definition 2.5.** *The **limit space** is the pair $(\overline{V_\infty}, \mathrm{G}_\infty)$ where $\overline{V_\infty}$ denotes the completion of $V_\infty$ with respect to $\|\cdot\|_{V_\infty}$,*

*endowed with the symmetrized metric*

$$\overline{\mathrm{d}}(x, y) := \inf_{g \in \mathrm{G}_\infty} \|g \cdot x - y\|_{V_\infty} \quad \textit{for } x, y \in \overline{V_\infty}. \quad (1)$$

*The **orbit space** is the metric space $(\overline{V_\infty}/\mathrm{G}_\infty, \overline{\mathrm{d}})$ with*

$$\overline{V_\infty}/\mathrm{G}_\infty := \overline{\{\mathrm{G}_\infty \cdot x \mid x \in \overline{V_\infty}\}},$$

*where the outer completion is taken with respect to $\overline{\mathrm{d}}$.*

This symmetrized metric is a pseudometric on $\overline{V_\infty}$ and a metric on the space of orbit closures of $\overline{V_\infty}$ under the action of $G_\infty$ (Levin et al., 2025). We slightly abuse notation, by denoting this set $\overline{V_\infty}/G_\infty$ and calling it "orbit space." Two elements $x, y \in \overline{V_\infty}$ are in the same orbit (closure) if, and only if, $\overline{\mathrm{d}}(x, y) = 0$. Let us provide a quick example that will be used recurrently.

**Example 2.6** (Duplication sequence with $\ell_p$ norms)**.** *Consider the duplication embedding from Example 2.2. Endow each $V_n = \mathbb{R}^n$ with the normalized $\ell_p$ norm for $p \in [1, \infty)$, i.e., $\|x\|_{V_n} = (\frac{1}{n} \sum_i^n x_i^p)^{1/p}$. Then, the limit space $V_\infty$ can be identified with $L^p([0, 1])$, with permutations $\mathrm{S}_n$ acting by permuting the $n$ consecutive intervals of length $1/n$ in the domain $[0, 1]$ of functions in $L^p([0, 1])$. The orbit space $\overline{V_\infty}/G_\infty$ corresponds to the space of probability distributions $\mathcal{P}_p(\mathbb{R})$. Furthermore, the symmetrized distance $\overline{\mathrm{d}}$ coincides with the Wasserstein-$p$ metric. Details appear in Appendix C.1.2.*

Armed with these notions, we are now ready to define continuity across dimensions.

**Definition 2.7.** *Let $\mathbb{V}$ be a consistent sequence endowed with a compatible invariant norm $(\|\cdot\|_{V_n})$. A sequence of invariant functions $(f_n : V_n \to \mathbb{R})$ is **continuously transferable** if $(i)$ there exists an extension $f : \overline{V_\infty} \to \mathbb{R}$, i.e., $f_n = f_\infty|_{V_n}$ for all $n$, and $(ii)$ the extension is continuous with respect to $\|\cdot\|_{V_\infty}$.*

We remark that if $(f_n)$ is continuously transferable, then it must be compatible. Further, an extension $f_\infty$ is continuous in the limit space with respect to $\|\cdot\|_{V_\infty}$ if, and only if, it is continuous in the orbit space $\overline{V_\infty}/G_\infty$ with respect to $\overline{\mathrm{d}}$; see Lemma A.4 in Appendix A. The term 'transferable' is motivated by the results in (Levin et al., 2025), which show that if a high-dimensional model is transferable, then it can be trained on low-dimensional inputs and generalize to higher-dimensional ones.

## 3. How to establish any-dimensional invariant universality?

In this section, we outline a general recipe for establishing any-dimensional invariant universality. We begin by formally stating our goal. Consider a continuously transferable, consistent sequence of invariant functions $(f_n : V_n \to \mathbb{R})$. Building upon the foundational concepts from Section 2,

our primary focus is on approximating its $G_\infty$-invariant extension $f_\infty$. To approximate this function we consider a class of continuously transferable any-dimensional models $\mathcal{F}$. We identify each of these models with its limiting extension $\widehat{f}_\infty$ and our goal is to show that: for any fixed accuracy $\varepsilon > 0$ and compact set $K$ in the domain of $f_\infty$, there exists $\widehat{f}_\infty \in \mathcal{F}$ such that

$$\sup_{x \in K} |f_\infty(x) - \widehat{f}_\infty(x)| \le \varepsilon. \quad (2)$$

We make two remarks concerning domains and topology. **Domain of the extension.** Since invariant functions are constant on orbits, they admit two equivalent viewpoints: as functions on the limit space $f_\infty : \overline{V_\infty} \to \mathbb{R}$, or as functions on the orbit space $f_\infty : \overline{V_\infty}/G_\infty \to \mathbb{R}$. Thus, the question of approximation can be posed over any of these domains. Any consistent norm $\|\cdot\|_{V_\infty}$ induces a topology over the limit space, the corresponding topology on the orbit space is given by the symmetrized metric (1). As it turns out, compact sets in the orbit space are 'richer' than those in the limit space. The underlying reason for this phenomenon is that by taking orbits, we collapse 'big' subsets (potentially noncompact) into single orbits. To illustrate this point, consider the setting in Example 2.6 where $\overline{V_\infty} = L^p([0, 1])$ are integrable functions and $\overline{V_\infty}/G_\infty = \mathcal{P}_p(\mathbb{R})$ are probability distributions with $p$th moments. While the set $\widetilde{K} = \{X \in L^p([0, 1]) \mid \mathrm{range}(X) \subseteq [0, 1]\}$ is not compact in $L^p([0, 1])$, the collection of its equivalence classes $K = \{\mu \in \mathcal{P}_p(\mathbb{R}) \mid \mathrm{supp}(\mu) \subseteq [0, 1]\}$ is compact in $\mathcal{P}_p(\mathbb{R})$; we defer details to Appendix B. In Section 4, we will see more examples of this phenomena. In what follows, we focus on universality on the orbit space equipped with the symmetrized metric, i.e., we prove (2) for compact subsets $K$ of the orbit space.

### Induced topology and activation functions.

In finite dimensional settings, all norms induce equivalent topologies, which means that continuity with respect to one norm guarantees continuity under any other. In sharp contrast, the topology of the limit space $\overline{V_\infty}$ (and correspondingly that of the orbit space) is heavily dependent on the choice of the consistent norm $\|\cdot\|_{V_\infty}$. This choice is often dictated by the learning task, i.e., the map $f_\infty$ we aim to approximate (or in practice learn) is naturally continuous with respect to certain norms but not others. To illustrate this point, consider again the setting in Example 2.6 with $p = 2$, and the sequence of 'second moment' functions

$$f_n(x) = \frac{1}{n} \sum_{j=1}^n x_j^2 \text{ with extension } f_\infty(\mu_X) = \mathbb{E}_{Y \sim \mu_X}[Y^2]$$

where $\mu_X \in \mathcal{P}_2(\mathbb{R})$ denotes the probability law associated with $X \in L^2([0, 1])$. In this case, $f_\infty$ is continuous with respect to $W_2$, but discontinuous with respect to $W_1$. As we will see in Section 4, several existing any-dimensional

architectures are not continuous in the natural topologies induced by norms of interest. A recurring theme is that the chosen topology constrains which activation functions and architectural primitives can be used.

**Desiderata and a recipe.** We now outline our strategy for proving universality. Our proof strategy is based on the Stone–Weierstrass theorem, which we recall for convenience; see (Rudin, 1991)[Theorem 5.7] for a proof.

**Theorem 3.1** (Stone-Weierstrass)**.** *Let* $C(K)$ *be the set of continuous real-valued functions on a compact metric space* $K$ *endowed with the norm* $\|f\|_\infty = \sup_{x \in K} |f(x)|$. *Let* $\mathcal{F}$ *be a subalgebra of* $C(K)$ *which contains a non-zero constant function.*[3] *Then* $\mathcal{F}$ *is dense in* $C(K)$ *if, and only if, it separates points.*[4]

Density in the sup-norm is equivalent to (2). This discussion suggests two desiderata for practical model classes $\mathcal{F}$: $(i)$ functions in $\mathcal{F}$ should be continuous with respect to the natural topology induced by the learning task; and $(ii)$ the class $\mathcal{F}$ should form a subalgebra that separates points. Somewhat surprisingly, many existing architectures fail to meet one (or both) of these requirements, likely because most prior work studies approximation on a fixed dimension $V_n$ rather than on the orbit space. Guided by these desiderata, we propose the following three-step recipe.

**Step 1 (Compact sets):** Given a consistent norm $\| \cdot \|_\infty$ dictated by the learning task, characterize compact sets in the orbit space $\overline{V_\infty}/G_\infty$.

**Step 2 (Continuity):** Construct a collection $\mathcal{F}$ of invariant any-dimensional models that are continuously transferable with respect to the symmetrized metric (1) on the orbit space $\overline{V_\infty}/G_\infty$.

**Step 3 (Universality):** Establish universality of $\mathcal{F}$ via the Stone–Weierstrass theorem by showing that $\mathcal{F}$ forms a subalgebra that separates points.

# 4. Instantiations of the recipe

In this section, we instantiate our general recipe for any-dimensional models on three representative domains: sets (Section 4.1), graphs (Section 4.2), and point clouds (Section 4.3). For brevity, we work directly with the corresponding infinite-dimensional extensions of our models, rather than the underlying sequences. Across all domains, we find that standard architectures are either discontinuous, or fail to be universal. We therefore either introduce minor, principled modifications to these existing architectures or propose new architectures, and we prove that the resulting models

---

[3]A subalgebra of functions is a set closed under addition, scalar multiplication, and products.

[4]The function class $\mathcal{F}$ separates points on $K$ if for all $x, y \in K$ and $x \neq y$, there is an $f \in \mathcal{F}$ such that $f(x) \neq f(y)$.

are both continuous and universal.

## 4.1. Set functions

We start with models for sets of any size. Since fully connected feedforward neural networks appear repeatedly throughout, we define the family

$$\mathrm{NN}_{k,m}^\phi := \bigcup_{r=1}^\infty \left\{ L_2 \circ \phi \circ L_1 \mid L_1 \in \mathcal{L}_{r,m}, L_2 \in \mathcal{L}_{k,r} \right\}, \quad (3)$$

where $\mathcal{L}_{m,n} := \{x \mapsto Wx + \theta \mid W \in \mathbb{R}^{m \times n}, \theta \in \mathbb{R}^m\}$ is the collection of affine maps from $\mathbb{R}^n$ to $\mathbb{R}^m$ and $\phi \colon \mathbb{R} \to \mathbb{R}$ is an activation function applied component-wise. Observe that these networks can be arbitrarily wide.

We consider two models whose input spaces parallel the consistent sequences in Example 2.2. Although both sequences share the same finite-dimensional spaces $V_n = \mathbb{R}^{n \times k}$, they use different embeddings and consistent norms, and therefore induce substantially different orbit spaces. In both settings we start from the DeepSets architecture (Zaheer et al., 2017), which maps

$$\mathrm{DeepSets}^{\rho,\sigma}(X) = \sigma \left( \sum_{i=1}^\infty \rho(X_{i:}) \right), \quad (4)$$

where $\rho \in \mathrm{NN}_{r,k}^\phi$, $\sigma \in \mathrm{NN}_{1,r}^\phi$, $r \in \mathbb{N}$. For each of the two orbit spaces we consider, we introduce a mild modification of DeepSets that restores continuity and yields universality.

### 4.1.1. UNIVERSALITY OVER SEQUENCES

Additional details about the following constructions can be found in Appendix C.1.1. For the first example, we consider the $k$-fold direct sum of the zero-padding consistent sequence $\mathbb{V}_{\mathrm{zero}}^{\oplus k} = \{(\mathbb{R}^{n \times k}), (\varphi_{N,n}^{\oplus k}), (\mathrm{S}_n)\}$. Given an arbitrary norm $\| \cdot \|_{\mathbb{R}^k}$ in $\mathbb{R}^k$, and a real number $p \in [1, \infty)$, we endow $\mathbb{V}_{\mathrm{zero}}^{\oplus k}$ with the compatible sequence of $\ell_p$ norm $\|X\|_{V_n} = (\sum_{i=1}^n \|X_{i:}\|_{\mathbb{R}^k}^p)^{1/p}$, which are unchanged under zero-padding. The resulting limit space is the space of sequences $\overline{V_\infty} = \ell_p(\mathbb{R}^k)$, with the group $G_\infty$ acting by permuting finitely many entries of these sequences.

**Step 1 (Compact sets).** For this example, we consider orbit closures of compact sets $K \subseteq \ell_p(\mathbb{R}^k)$, which we denote by $K/G_\infty$. These are compact subsets of $\overline{V_\infty}/G_\infty$ by the definition of the quotient topology on $\overline{V_\infty}/G_\infty$. Compact sets in $\ell_p(\mathbb{R}^k)$ are well-understood.

**Proposition 4.1** (Compactness in $\ell_p(\mathbb{R}^k)$, (Diestel, 2012))**.** *For* $p \in [1, \infty)$, *a set* $K \subseteq \ell_p(\mathbb{R}^k)$ *is compact if, and only if, it satisfies that* $(i)$ *it is closed;* $(ii)$ *it is bounded, i.e.,* $\sup_{X \in K} \|X\|_p < \infty$; *and* $(iii)$ *it exhibits uniform tail decay, i.e.,* $\lim_{N \to \infty} \sup_{X \in K} (\sum_{i \geq N} \|X_{i:}\|_{\mathbb{R}^k}^p)^{1/p} = 0$.

For instance, the set $\{(x_i)_{i=1}^\infty : |x_i| \leq 2^{-i} \text{ for all } i \in \mathbb{N}\}$ is compact in $\ell_p(\mathbb{R})$ for all $p \in [1, \infty)$. We now turn to

constructing a family of continuous models on $\overline{V_\infty}/G_\infty$.

**Step 2 (Continuity).** Our architecture is based on DeepSets (4). To ensure that our architecture is compatible with respect to our consistent sequence structure, i.e., that it is unchanged under zero-padding, we assume that $\rho\left(\mathbf{0}_k\right) = \mathbf{0}_r$. In general, (4) is not a continuous model as the infinite sum may diverge since the function $\rho$ can decay arbitrarily slowly. Motivated by this observation, we propose

$$\text{DeepSets}_\infty^{\rho;\sigma}(X) = \sigma\left(\sum_{i=1}^\infty \|X_{i:}\|_{\mathbb{R}^k}^p \cdot \rho(X_{i:})\right), \quad (5)$$

where $\rho \in \text{NN}_{r,k}^\phi$ and $\sigma \in \text{NN}_{1,r}^\phi$ with $r \in \mathbb{N}$. The following proposition shows that $\text{DeepSets}_\infty^{\rho;\sigma}$ is indeed continuous on the space $\overline{V_\infty}/G_\infty$ of orbit closures when both $\rho$ and $\sigma$ are continuous. The proof is deferred to Appendix C.2.

**Theorem 4.2** (Continuity on sequences). *Suppose $\rho \in C(\mathbb{R}^k, \mathbb{R}^r)$ and $\sigma \in C(\mathbb{R}^r, \mathbb{R})$ are continuous functions. Then, the map $\text{DeepSets}_\infty^{\rho;\sigma}: \overline{V_\infty}/G_\infty \to \mathbb{R}$ defined in (5) is continuous with respect to the symmetrized metric (1).*

As an immediate consequence, we get that all functions in

$$\mathcal{F}_{\text{DS}} := \left\{\text{DeepSets}_\infty^{\rho;\sigma} \mid r \in \mathbb{N}, \rho \in \text{NN}_{r,k}^\phi, \sigma \in \text{NN}_{1,r}^\phi\right\}$$

are continuous.

**Step 3 (Universality).** Finally, we prove that the above variation of DeepSets is universal for sequences. The proof of the following result is deferred to Appendix C.3.

**Theorem 4.3** (Universality of $\text{DeepSets}_\infty$). *Suppose that the activation $\phi$ is continuous and non-polynomial. Let $K \subseteq \ell_p(\mathbb{R}^k)$ be any compact set. Then, $\mathcal{F}_{\text{DS}}$ is dense in $C(K/G_\infty)$.*

### 4.1.2. Universality over measures

Additional details about the constructions in this section are deferred to Appendix C.1.2. We consider the $k$-fold direct sum of the duplication embedding consistent sequence $\mathbb{V}_{\text{dup}}^{\oplus k} := \{(\mathbb{R}^{n\times k}), (\varphi_{N,n}^{\oplus k}), (\text{S}_n)\}$ defined as in Example 2.2. Given an arbitrary norm $\|\cdot\|_{\mathbb{R}^k}$ on $\mathbb{R}^k$, and a real number $p \in [1, \infty)$, we endow each space $\mathbb{R}^{n\times k}$ with the normalized $\ell_p$ norm, i.e., $\|X\|_{\bar{p}} = (\frac{1}{n}\sum_{i=1}^n \|X_{i:}\|_{\mathbb{R}^k}^p)^{1/p}$. As explained in Example 2.6, the resulting limit space is the space of functions $L^p([0,1], \mathbb{R}^k)$, and the space of orbit closures $\overline{V_\infty}/G_\infty$ can be identified with the Wasserstein space $\mathcal{P}_p(\mathbb{R}^k)$ equipped with the Wasserstein-$p$ distance $W_p$.

**Step 1 (Compact sets).** We begin with a characterization of compact sets in the orbit space.

**Proposition 4.4** (Compacta in $\mathcal{P}_p(\mathbb{R}^k)$ (Ambrosio et al., 2005)). *For $p \in [1, \infty)$, a subset $Q \subseteq \mathcal{P}_p(\mathbb{R}^k)$ is compact if, and only if, it is (i) closed; (ii) tight, i.e., for any $\varepsilon > 0$, there is a compact $K_\varepsilon \subseteq \mathbb{R}^k$ such that $\sup_{\mu\in Q} \mu\left(\mathbb{R}^k\backslash K_\varepsilon\right) \leq \varepsilon$; and (iii) p-uniformly integrable,*

*i.e., $\lim_{R\to\infty} \sup_{\mu\in Q} \int_{\|x\|_{\mathbb{R}^k}>R} \|x\|_{\mathbb{R}^k}^p \, \mathrm{d}\mu(x) = 0$.*

For example, the collection of all measures supported on a closed ball of radius $R > 0$, $Q = \{\mu \in \mathcal{P}_p(\mathbb{R}^k) \mid \text{supp}(\mu) \subseteq \overline{B(0,R)}\}$, is compact in this space. We now turn to constructing a family of continuous models on $\mathcal{P}_p(\mathbb{R}^k)$ that can approximate any continuous function on a compact subset to arbitrary precision.

**Step 2 (Continuity).** We consider the normalized DeepSets architecture proposed in (Bueno & Hylton, 2021),

$$\overline{\text{DeepSets}}_\infty^{\rho,\sigma}(\mu) = \sigma\left(\int \rho \, d\mu\right), \quad (6)$$

where $\rho \in \text{NN}_{r,k}^\phi$ and $\sigma \in \text{NN}_{1,r}^\phi$ with $r \in \mathbb{N}$. To establish the continuity of this architecture we require that the entries of $\rho$ do not grow too fast. In particular, we denote by $\mathcal{F}_p(\mathbb{R}^k, \mathbb{R}^r)$ the class of continuous functions $\rho = (\rho_1, \ldots, \rho_r)^\top$ such that each component $\rho_j$ satisfies a $p$-th order growth condition; specifically, for each $j \in [r]$, there exists a constant $M_j > 0$ such that

$$|\rho_j(x)| \leq M_j(1 + \|x\|_{\mathbb{R}^k}^p), \quad \forall x \in \mathbb{R}^k.$$

The following theorem establishes the continuity of $\overline{\text{DeepSets}}_\infty^{\rho,\sigma}$ under this growth condition. We defer its proof to Appendix C.4.

**Theorem 4.5** (Continuity on probability measures). *Suppose $\rho \in \mathcal{F}_p(\mathbb{R}^k, \mathbb{R}^r)$ and $\sigma \in C(\mathbb{R}^r, \mathbb{R})$ are continuous functions. Then, the map $\overline{\text{DeepSets}}_\infty^{\rho,\sigma}: \mathcal{P}_p(\mathbb{R}^k) \to \mathbb{R}$, defined in (6), is continuous with respect to $W_p$.*

As an immediate consequence of this theorem, we obtain that all functions in

$$\mathcal{F}_{\overline{\text{DS}}} := \left\{\overline{\text{DeepSets}}_\infty^{\rho,\sigma} \mid r \in \mathbb{N}, \rho \in \text{NN}_{r,k}^\phi, \sigma \in \text{NN}_{1,r}^\phi\right\}$$

are continuous provided that $\phi$ is Lipschitz, since it ensures that $\text{NN}_{r,k}^\phi \subseteq \mathcal{F}_p(\mathbb{R}^k, \mathbb{R}^r)$, for $p \in [1, \infty)$.

**Step 3 (Universality).** Finally, we establish the universality of the normalized DeepSets architecture in (6). We defer the proof of this result to Appendix C.5.

**Theorem 4.6** (Universality of $\overline{\text{DeepSets}}_\infty$). *Suppose that $\phi$ is a Lipschitz continuous, non-polynomial activation that is asymptotically polynomial at $\pm\infty$.[5] Let $Q \subseteq \mathcal{P}_p(\mathbb{R}^k)$ be a compact set. Then, $\mathcal{F}_{\overline{\text{DS}}}$ is dense in $C(Q)$.*

The special case of Theorem 4.6 with $Q$ consisting of all measures supported on a fixed compact set was proved in (Bueno & Hylton, 2021). The requirement that $\phi$ is asymptotically polynomial at $\pm\infty$ guarantees the universality of neural networks on non-compact domains (van Nuland, 2024). Representative activation functions meet-

---

[5]We say that $\phi(t)$ is asymptotically polynomial at $\pm\infty$ if there exist polynomials $P_+$ and $P_-$ (possibly constant) such that $\phi(t)/P_\pm(t) \to 1$ as $t \to \pm\infty$, respectively.

ing these conditions include ReLU, Softplus, and Sigmoid, among others.

## 4.2. Graph functions

Additional details about the constructions in this section can be found in Appendix D.1. Recall the divisibility ordering from the duplication embedding in Example 2.2. We consider a consistent sequence indexed with this ordering and given by $\mathbb{V}_{\text{dup}}^G = \{(\mathbb{R}_{\text{sym}}^{n\times n}), (\varphi_{N,n}), (\mathsf{S}_n)\}$, where $\mathbb{R}_{\text{sym}}^{n\times n}$ denotes the space of $n \times n$ symmetric matrices with real entries, and $\varphi_{N,n}$ converts each entry into $N/n \times N/n$ blocks with the same entry. We endow $\mathbb{R}_{\text{sym}}^{n\times n}$ with the cut norm, $\|A\|_\square = \frac{1}{n^2} \max_{I,J\subseteq[n]} |\sum_{i\in I, j\in J} A_{ij}|$. The limit space $\overline{V_\infty}$ can be identified with the space of kernels $\mathcal{W}$, i.e., measurable symmetric functions $W\colon [0,1]^2 \to \mathbb{R}$ equipped with the cut norm $\|\cdot\|_\square$. The symmetrized metric (1) corresponds to the so-called cut metric $\delta_\square$. This limit space has been widely studied in the literature on large graphs or graphons (Lovász, 2012). Intuitively, the functions $W$ model adjacency matrices of infinite-node graphs.

**Step 1 (Compact sets).** As is standard in the graphon literature, we consider graphs with bounded edge weights; in particular, we restrict them to the unit interval $[0,1]$. With this restriction, we focus on the subset of orbits corresponding to graphons, that is,

$$\mathcal{W}_0 := \{W \in \mathcal{W} \mid \text{range}(W) \subseteq [0,1]\} / \sim, \quad (7)$$

where $W_1 \sim W_2$ whenever $\delta_\square(W_1, W_2) = 0$ (Lovász, 2012). This is the collection of all limits of simple graphs, i.e., graphs with edge weights in $\{0,1\}$. This entire graphon space is compact.

**Proposition 4.7** (Compactness of $\mathcal{W}_0$, (Lovász, 2012)). *The space $\mathcal{W}_0$ endowed with the cut metric $\delta_\square$ is compact.*

**Step 2 (Continuity).** Achieving continuity in the cut metric is subtle. In particular, pointwise nonlinearities are not continuous in cut metric: for any pointwise map $\phi$, the operator $W \mapsto \phi(W)$ is continuous with respect to $\|\cdot\|_\square$ if and only if $\phi$ is linear (Herbst & Jegelka, 2025). Since continuity with respect to $\|\cdot\|_\square$ on the limit space is equivalent to continuity with respect to $\delta_\square$ on the orbit space (Lemma A.4), this precludes using pointwise activations. As it turns out, many existing graph architectures are discontinuous in $(\mathcal{W}_0, \delta_\square)$ (Maron et al., 2019b; Keriven & Peyré, 2019). One way to circumvent this issue is to coarsen the topology to make pointwise nonlinearities continuous, as in (Herbst & Jegelka, 2025). In contrast, we work directly with the cut distance $\delta_\square$, which metrizes a standard notion of graph limits (Lovász, 2012).

The continuity with respect to this distance is closely tied to *homomorphism densities*. Intuitively, these are functionals that quantify how frequently a fixed motif (a finite graph)

appears in a graphon. Formally, given a motif $F = (V, E)$ and a graphon $W \in \mathcal{W}_0$, the homomorphism density is

$$t(F, W) = \int_{[0,1]^{|V|}} \prod_{ij\in E} W(x_i, x_j) \prod_{i\in V} dx_i. \quad (8)$$

In turn, the linear span of homomorphism densities of simple graphs, $\mathcal{HD} := \text{span}\{t(F, \cdot) \mid F \text{ is simple}\}$, is dense in the space of continuous functions $C(\mathcal{W}_0, \delta_\square)$ (Lovász, 2012). Thus, it suffices to find a model class that can be expressed as linear combinations of homomorphism densities of simple graphs. Importantly, the desired model class must also not contain any homomorphism densities of non-simple graphs, as these are discontinuous in the cut metric (Lovász, 2012). With these considerations in mind, we propose the following family of models

$$h_m^{a,b}(W) = \int_{[0,1]^m} \prod_{1\le i<j\le m} (a_{ij}W(x_i,x_j) + b_{ij}) \prod_{\ell=1}^m dx_\ell, \quad (9)$$

where $a_{ij}, b_{ij} \in \mathbb{R}$ are parameters. In particular, the restriction $i < j$ ensures that only simple motifs can be parameterized. Based on this, we define the function class

$$\mathcal{F}_\mathcal{W} := \text{span}\{h_m^{a,b}(\cdot) \mid m \in \mathbb{N}, \, m \ge 2, \, a, b \in \mathbb{R}_{\text{sym}}^{m\times m}\}. \quad (10)$$

As we detail in Appendix D.4, this family can be represented by a neural-network-like deep architecture, making it directly amenable to gradient-based optimization. This property is informally stated as follows.

**Theorem 4.8** (Informal). *The family of functions $\mathcal{F}_\mathcal{W}$ can be parameterized using a neural network architecture.*

The neural network architecture in this result is somewhat involved; to streamline the presentation, we defer to Appendix D.4. We mention in passing that the nonlinearities correspond to tensor contractions, which have been proposed in the past to enhance the expressivity of invariant architectures (Finzi et al., 2021). Our discussion thus far yields the following result; its formal proof appears in Appendix D.2.

**Theorem 4.9** (Continuity of $\mathcal{F}_\mathcal{W}$). *All functions in $\mathcal{F}_\mathcal{W}$ are continuous with respect to $\delta_\square$.*

**Step 3 (Universality).** Separating points in the graphon space $(\mathcal{W}_0, \delta_\square)$ corresponds to distinguishing graphs up to weak isomorphism—a fundamental task in the study of graph neural networks. Prior work on the expressive power of standard GNN architectures (Xu et al., 2018; Maron et al., 2019a; Morris et al., 2019) has linked their expressivity to the Weisfeiler–Lehman (WL) graph isomorphism test (Leman & Weisfeiler, 1968), which fails to distinguish certain non-isomorphic graphs. In contrast, the following theorem establishes that our model class (10) is universal; its proof is deferred to Appendix D.3.

**Theorem 4.10** (Universality of $\mathcal{F}_{\mathcal{W}}$). *The class of functions $\mathcal{F}_{\mathcal{W}}$ in* (10) *is dense in* $C(\mathcal{W}_0, \delta_\square)$.

The proof of Theorems 4.9 and 4.10 essentially proceed by expanding out the product in the definition of our model (9) and observing that it is a linear combination of homomorphism densities. By appropriately choosing the parameters $a, b$ in (9), we further show that all homomorphism densities lie within our parametric family $\mathcal{F}_{\mathcal{W}}$.

We emphasize at this point that there are two key advantages to working with the parametrization (9) over directly learning coefficients in a linear combination of homomorphism densities, as done in (Maehara & NT, 2019; Nguyen & Maehara, 2020) for example. The first advantage pertains to the number of parameters needed to represent all homomorphism densities of simple graphs on $m$ nodes. While the number of such densities scales as $\mathcal{O}\left(2^{\binom{m}{2}}/m!\right)$ and these densities are all linearly independent, our model is able to represent all of them using only $m(m-1)$ parameters. The second advantage pertains to the implementation and training of our model. Learning coefficients in a linear combination of homomorphism densities requires computing each homomorphism density separately for every motif and every input graph, which is costly. Our parameterization avoids this by encoding graph patterns implicitly through the learnable coefficients $a, b$ in (9). Moreover, our model is directly amenable to gradient-based optimization thanks to its neural network-like decomposition in Appendix D.4.

The key enabler of universality here is the use of high-order tensors as hidden layers; the order of these tensors—as well as the width and depth of our networks—grows with $m$. Whether universal approximation is achievable with fixed width or depth remains an interesting open question.

### 4.3. Point cloud functions

Details of the constructions in this section appear in Appendix E.1. We consider a consistent sequence analogous to the one used for sets (measures), namely $\mathbb{V}_{\text{dup}}^P = \{(\mathbb{R}^{n \times k}), (\varphi_{N,n}^{\oplus k}), (\mathrm{S}_n \times \mathrm{O}(k))\}$, where $\mathbb{R}^{n \times k}$ represents sets of $n$ points in $\mathbb{R}^k$ for fixed $k \in \mathbb{N}$, and the maps $\varphi_{N,n}$ are duplication embeddings. The key difference is that the group is now $\mathrm{S}_n \times \mathrm{O}(k)$, where $\mathrm{S}_n$ is the permutation group and $\mathrm{O}(k)$ is the orthogonal group, acting via $(g, h) \cdot X = gXh^\top$. We endow $\mathbb{R}^k$ with the Euclidean norm $\|\cdot\|_{\mathbb{R}^k}$, and equip each $V_n$ with the normalized $\ell_2$ norm; the limit space can then be identified with $\overline{V_\infty} = L^2([0,1]; \mathbb{R}^k)$.

Since equivariant operators appear repeatedly throughout, we define the family

$$\mathrm{LE}_{k \to l} := \left\{ L \in \mathbb{B}(k, l) \mid L \text{ is } S_{[0,1]}\text{-equivariant} \right\}, \quad (11)$$

where $\mathbb{B}(k, l) = \mathcal{B}\left(L^2([0,1]^k), L^2([0,1]^l)\right)$ denotes the

space of bounded linear operators and $S_{[0,1]}$ is the group of measure-preserving bijections $\varphi: [0,1] \to [0,1]$. We say that $L$ is $S_{[0,1]}$-equivariant if, for every $\varphi \in S_{[0,1]}$ and every $W \in L^2([0,1]^k)$, one has $L(W^\varphi) = L(W)^\varphi$ almost everywhere, where $W^\varphi(x_1, \ldots, x_k) := W(\varphi(x_1), \ldots, \varphi(x_k))$.

**Step 1 (Compact sets).** For this setting, we consider the set of orbit closures arising from a set of bounded functions in the limit space $\overline{V_\infty}$, namely

$$K_R := \{X \in L^2\left([0,1]; \mathbb{R}^k\right) \mid \|X\|_{L^\infty} \le R\},$$

where $R > 0$ is a positive constant. The next proposition establishes that the space of orbit closures of $K_R$, which we denote by $K_R/G_\infty$, is compact in $\overline{V_\infty}/G_\infty$. The proof is deferred to Appendix E.2.

**Proposition 4.11** (Compactness of $K_R/G_\infty$). *The set of orbit closures $K_R/G_\infty$ is a compact subset of $\overline{V_\infty}/G_\infty$.*

**Step 2 (Continuity).** Next, we introduce a class of continuous models on the space of orbit closures. The architecture decomposes into two stages: given a point cloud $X$, we first form its Gram (inner-product) matrix $X^\top X$, and then apply a graph neural network (GNN) to this matrix. Intuitively, the Gram map enforces invariance to $\mathrm{O}(k)$, while the GNN provides invariance to permutations $S_n$. Closely related architectures were first proposed in (Villar et al., 2021).

Let us formally define the architecture in the orbit space. For the first stage, we map each $X \in K_R/G_\infty$ to a graphon $W_X: [0,1]^2 \to [0,1]$ defined by

$$W_X(x, y) := \frac{1}{2R^2} \langle X(x), X(y) \rangle + \frac{1}{2}, \quad (12)$$

where we interpret $X$ as any representative in its equivalence class. Notice that $W_X$ is a symmetric measurable function, and so can be seen as a graphon. We endow the space of graphons with the $\delta_2$ metric given by

$$\delta_2(W, \widetilde{W}) := \inf_{\varphi \in S_{[0,1]}} \|W - \widetilde{W}^\varphi\|_2.$$

For the second stage, we use Invariant Graphon Networks (IGNs) (Herbst & Jegelka, 2025), i.e., functions $I_{\varrho, M, L, b}: \mathcal{W}_0 \to \mathbb{R}$ of the form

$$I_{\varrho, M, L, b}(W) := \sum_{m=1}^M L_m^{(2)}\left(\varrho\left(L_m^{(1)}(W) + b_m^{(1)}\right)\right) + b^{(2)}, \quad (13)$$

where $M \in \mathbb{N}_0$, $L_m^{(1)} \in \mathrm{LE}_{2 \to k_m}$, $L_m^{(2)} \in \mathrm{LE}_{k_m \to 0}$, $b_m^{(1)}, b^{(2)} \in \mathbb{R}$, and $k_m \in \mathbb{N}$ for each $m \in \{1, \ldots, M\}$. Here $\varrho: \mathbb{R} \to \mathbb{R}$ is a nonlinearity acting pointwise.

Altogether, we consider the family of models given by

$$\mathcal{F}_{PC}^\varrho := \{X \mapsto I_{\varrho, M, L, b}(W_X)\}. \quad (14)$$

Here the architectural hyperparameters and parameters (i.e., $M$, the operators $L$, and the biases $b$) range over the admissible sets described after (13); we omit the full specification

for brevity. The next theorem shows that this architecture is indeed continuous with respect to the symmetrized distance. The proof is in Appendix E.3.

**Theorem 4.12** (Continuity of $\mathcal{F}_{PC}^{\varrho}$)**.** *Let $\varrho$ be continuous. Then, all models in $\mathcal{F}_{PC}^{\varrho}$ are continuous with respect to the symmetrized metric.*

**Step 3 (Universality).** To close, we show that our model class for point clouds (14) is universal; the proof is deferred to Appendix E.4.

**Theorem 4.13** (Universality of $\mathcal{F}_{PC}^{\varrho}$)**.** *Let $\varrho$ be continuous and non-polynomial. Then, $\mathcal{F}_{PC}^{\varrho}$ is dense in $C\left(K_R/G_{\infty}\right)$.*

The above theorem gives universality in the space of continuous functions of point clouds that are invariant under rotations of the cloud. If we center the input point cloud before applying our proposed architecture, we also obtain universality in the smaller space of continuous functions invariant under all rigid motions of their inputs.

## 5. Conclusions and future work

To summarize, we have (i) formalized universality for any-dimensional invariant architectures, (ii) proposed a general recipe for constructing and certifying any-dimensional universal model classes, and (iii) used this recipe to prove universality for several concrete families. A key limitation of our framework is that it applies only to scalar-valued outputs. Many practically relevant any-dimensional architectures have both input and output sizes growing and are naturally equivariant as opposed to invariant. In this setting, the problem no longer reduces to studying functionals on orbit spaces. Developing tools that exploit such equivariance to obtain universality results for more general architectures is an important direction for future work. Another limitation is that we only focus on dense graphs in our current instantiations. There are several notions of limits for sparse graphs including (Borgs et al., 2019; 2018; Backhausz & Szegedy, 2022), and applying our framework to these limits is another interesting direction.

## Impact Statement

This paper presents work whose goal is to advance the field of Machine Learning. Our work is purely theoretical and we do not expect it would have any societal consequences.

## Acknowledgments

MD was partially supported by NSF awards CCF 2442615 and DMS 2502377, and a Sloan Research Fellowship. EL was partially supported by AFOSR FA9550-23-1-0070 and FA9550-23-1-0204

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

# A. Missing details from Section 2

In this section, we present missing details from Section 2. We start with a more detailed definition of a consistent sequence.

**Definition A.1** (Detailed version of Def. 2.1). *A **consistent sequence** of group representations over directed poset $(\mathbb{N}, \preceq)$ is a triple $\mathbb{V} = \{(V_n)_{n \in \mathbb{N}}, (\varphi_{N,n})_{n \preceq N}, (G_n)_{n \in \mathbb{N}}\}$ consisting of the following elements.*

1. *(**Groups**) A sequence of groups $(G_n)$ indexed by $\mathbb{N}$ that embed into each other. Specifically, whenever $n \preceq N$ there is an injective group homomorphism $\theta_{N,n} \colon G_n \to G_N$ with*

$$\theta_{i,i} = \mathrm{id}_{G_i} \quad \text{for all } i \in \mathbb{N},$$
$$\theta_{k,j} \circ \theta_{j,i} = \theta_{k,i} \quad \text{whenever } i \preceq j \preceq k \text{ in } \mathbb{N}.$$

2. *(**Vector spaces**) A sequence of finite-dimensional, real vector spaces $(V_n)$ indexed by $\mathbb{N}$ such that each $V_n$ is a $G_n$-representation.*

3. *(**Embeddings**) A collection of embeddings $(\varphi_{N,n} \colon V_n \hookrightarrow V_N)_{n \preceq N}$ such that $\varphi_{N,n}$ is $G_n$-equivariant, i.e.,*

$$\varphi_{N,n}(g \cdot v) = \theta_{N,n}(g) \cdot \varphi_{N,n}(v) \text{ for all } g \in G_n, v \in V_n.$$

*and such that*

$$\varphi_{i,i} = \mathrm{id}_{V_i} \quad \text{for all } i \in \mathbb{N},$$
$$\varphi_{k,j} \circ \varphi_{j,i} = \varphi_{k,i} \quad \text{whenever } i \preceq j \preceq k \text{ in } \mathbb{N}.$$

We can take direct sums of consistent sequences to obtain richer consistent sequences, as done in several of the examples of Section 4.

**Definition A.2.** *The $k$-fold direct sum of $\mathbb{V}$ is defined as*

$$\mathbb{V}^{\oplus k} := \{(V_n^{\oplus k}), (\varphi_{N,n}^{\oplus k}), (G_n)\}$$

*where $V_n^{\oplus d}$ denotes the direct sum of $d$ copies of $V_n$ and $\varphi_{N,n}^{\oplus d} \colon V_n^{\oplus d} \to V_N^{\oplus d}$ is defined by applying $\varphi_{N,n}$ to each component. The group $G_n$ acts on $V_n^{\oplus d}$ by simultaneously acting on every copy of $V_n$, i.e. $g \cdot (v_1, \ldots, v_d) := (g \cdot v_1, \ldots, g \cdot v_d)$.*

The next two propositions show that compatibility (Definition 2.4) is equivalent to the existence of a limiting extension.

**Proposition A.3** ((Levin et al., 2025, Prop. B.6)). *Let $\mathbb{V} = \{(V_n), (\varphi_{N,n}), (G_n)\}$ and $\mathbb{U} = \{(U_n), (\psi_{N,n}), (G_n)\}$ be consistent sequences. A sequence of maps $(f_n \colon V_n \to U_n)$ is compatible if, and only if, it extends to the limit, i.e., there exists a $G_\infty$-equivariant map $f_\infty \colon V_\infty \to U_\infty$ such that $f_n = f_\infty|_{V_n}$ for all $n$.*

In particular, a sequence of norms $\|\cdot\|_{V_n}$ is compatible if and only if they extend to a norm $\|\cdot\|_{V_\infty}$ preserved by the action of $G_\infty$.

Finally, we show that continuity of invariant functions in the limit space is equivalent to continuity of the corresponding induced function on the orbit space.

**Lemma A.4.** *Let $\|\cdot\|_{V_\infty}$ be the extension of a compatible sequence of norms $\|\cdot\|_{V_n}$. Let $f_\infty \colon \overline{V_\infty} \to \mathbb{R}$ be $G_\infty$-invariant function, and let $\bar{f}_\infty$ be the induced function on the space of orbit closures $\overline{V_\infty}/G_\infty$. Then, $f_\infty$ is continuous on $\left(\overline{V_\infty}, \|\cdot\|_{V_\infty}\right)$ if, and only if, $\bar{f}_\infty$ is continuous on $\left(\overline{V_\infty}/G_\infty, \overline{\mathrm{d}}\right)$.*

*Proof of Lemma A.4.* Let $\pi \colon \overline{V_\infty} \to \overline{V_\infty}/G_\infty$ be the quotient map $\pi(x) = [x]$. Because $f_\infty$ is $G_\infty$-invariant, we can write $\bar{f}_\infty \circ \pi$ for some $\bar{f}_\infty \colon \overline{V_\infty}/G_\infty \to \mathbb{R}$. We prove the two implications.

($\Leftarrow$) Suppose $\bar{f}_\infty \colon \overline{V_\infty}/G_\infty \to \mathbb{R}$ is continuous with respect to $\overline{\mathrm{d}}$. Note that $\pi$ is continuous with respect to the norm $\|\cdot\|_{V_\infty}$ on $\overline{V_\infty}$ and symmetrized metric $\overline{\mathrm{d}}$ on $\overline{V_\infty}/G_\infty$, since $\overline{\mathrm{d}}(\pi(x), \pi(y)) = \inf_{g \in G_\infty} \|g \cdot x - y\|_{V_\infty} \leq \|x - y\|_{V_\infty}$. Therefore, the composition $f_\infty = \bar{f}_\infty \circ \pi$ is continuous with respect to $\|\cdot\|_{V_\infty}$.

($\Rightarrow$) Suppose $f_\infty \colon \overline{V_\infty} \to \mathbb{R}$ is continuous with respect to $\|\cdot\|_{V_\infty}$. Then for any $\varepsilon > 0$ and any $[x] \in \overline{V_\infty}/G_\infty$ there exists $\delta > 0$ such that any $y \in \overline{V_\infty}$ with $\|x - y\|_{V_\infty} < \delta$ satisfies $|f_\infty(x) - f_\infty(y)| < \varepsilon$. Now, consider any $[y'] \in \overline{V_\infty}/G_\infty$ satisfying $\overline{\mathrm{d}}([x], [y']) < \delta$, then by the definition of $\overline{\mathrm{d}}$, there exists an element $g_0 \in G_\infty$ such that $\|g_0 \cdot x - y'\|_{V_\infty} < \delta$. Since

for any $g \in G_\infty$, the map $x \mapsto g \cdot x$ is an isometry on $\overline{V_\infty}$, we have $\|x - g_0^{-1} \cdot y'\|_{V_\infty} = \|g_0 \cdot x - y'\|_{V_\infty} < \delta$, which implies $|f_\infty(x) - f_\infty(g_0^{-1} \cdot y')| < \varepsilon$. Since $f_\infty$ is $G_\infty$-invariant, we obtain $|\bar{f}_\infty([x]) - \bar{f}_\infty([y'])| = |f_\infty(x) - f_\infty(g_0^{-1} \cdot y')| < \varepsilon$, and conclude that $\bar{f}_\infty$ is continuous. $\qquad\square$

## B. Missing details from Section 3

In this section, we establish the claim that the set of functions in $L^p[0, 1]$ with uniformly bounded range is not compact. This is a well known fact, but we prove it for completeness.

**Lemma B.1.** *The set* $\widetilde{K} = \{X \colon [0, 1] \to [0, 1]\}$ *is not compact in* $L^p([0, 1])$ *for any* $p \in [1, \infty]$.

*Proof of Lemma B.1.* It suffices to exhibit an infinite sequence $(X_n) \subseteq \widetilde{K}$ that has no convergent subsequence. To this end, let $X_n(t) = 1$ if $t \in \bigcup_{i=1}^{2^{n-1}} \left[\frac{2i-1}{2^n}, \frac{2i}{2^n}\right]$ and $X_n(t) = 0$ otherwise. Note that $\|X_n - X_m\|_p = \frac{1}{4^{1/p}}$ for all $n \neq m$, so $(X_n)$ cannot have a convergent subsequence in $L^p$.

$\qquad\square$

## C. Missing proofs and additional details from Section 4.1

In this section, we present the proofs and additional details about the constructions for set problems.

### C.1. Consistent sequences on sets

We start by elaborating on the consistent sequences from Section 4.1. These were studied in detail in (Levin et al., 2025, Appx. F), and we refer the reader there for more details and proofs.

#### C.1.1. ZERO-PADDING CONSISTENT SEQUENCE $\mathbb{V}_{\text{zero}}$ WITH $\ell_p$ NORM

The zero-padding consistent sequence $\mathbb{V}_{\text{zero}} = \{(V_n), (\varphi_{N,n}), (G_n)\}$ is defined as Example 2.2. Here we set $G_n = S_n$ to be the permutation group acting on $\mathbb{R}^n$ by permuting coordinates, i.e., $(g \cdot x)_i = x_{g^{-1}(i)}$ for $g \in S_n$. For $n \leq N$, the embedding of groups $\theta_{N,n} \colon S_n \to S_N$ is given by

$$\theta_{N,n}(g) = \left[\begin{array}{cc} g & 0 \\ 0 & I_{N-n} \end{array}\right] \quad \text{for } g \in S_n.$$

For $p \in [1, \infty)$, each $V_n$ is equipped with the $\ell_p$-norms $\|x\|_p = \left(\sum_{i=1}^n |x_i|^p\right)^{1/p}$ which are permutation-invariant. By Proposition A.3, this induces a norm on $V_\infty$, also denoted as $\|\cdot\|_p$. Consequently, the limit space is identified with the classical sequence space

$$\overline{V_\infty} = \ell_p = \left\{ x = (x_i)_{i=1}^\infty : \|x\|_p = \left(\sum_{i=1}^\infty |x_i|^p\right)^{1/p} < \infty \right\}.$$

The symmetrized distance $\overline{d}_p(x, y) = \inf_{g \in G_\infty} \|g \cdot x - y\|_p$ defines a metric on the space of orbit closures $\overline{V_\infty}/G_\infty$.

Next, consider the $k$-fold direct sum $\mathbb{V}_{\text{zero}}^{\oplus k} = \{(\mathbb{R}^{n \times k}), (\varphi_{N,n}^{\oplus k}), (S_n)\}$ of $\mathbb{V}_{\text{zero}}$, as defined in Definition A.2. Fix an arbitrary norm $\|\cdot\|_{\mathbb{R}^k}$ on $\mathbb{R}^k$, and define the $\ell_p$-norm on $\mathbb{R}^{n \times d}$ as

$$\|X\|_p = \left(\sum_{i=1}^n \|X_{i:}\|_{\mathbb{R}^k}^p\right)^{1/p},$$

where $X_{i:}$ denotes the $i$-th row of $X$. The corresponding limit space can be represented as:

$$\overline{V_\infty} = \ell_p(\mathbb{R}^k) = \left\{ X = (X_{i:})_{i=1}^\infty : \|X\|_p = \left(\sum_{i=1}^\infty \|X_{i:}\|_{\mathbb{R}^k}^p\right)^{1/p} < \infty \right\}.$$

C.1.2. DUPLICATION CONSISTENT SEQUENCE $\mathbb{V}_{\text{dup}}$ WITH NORMALIZED $\ell_p$-NORMS

The duplication embedding consistent sequence $\mathbb{V}_{\text{dup}} = \{(V_n), (\varphi_{N,n}), (G_n)\}$ is defined as Example 2.2. The group $G_n = S_n$ again acts on $\mathbb{R}^n$ by permuting coordinates. For $n|N$, the embedding of groups $\theta_{N,n} \colon S_n \to S_N$ is given by $\theta_{N,n}(g) = g \otimes I_{N/n}$, where $\otimes$ denotes the Kronecker product.

The space $V_\infty$ can be identified with the space of step functions, whose discontinuity points are rational. More precisely, each $x \in \mathbb{R}^n$ corresponds to a step function $f_x \colon [0,1] \to \mathbb{R}$ given by

$$f_x(t) = x_i \quad \text{for } t \in \left(\frac{i-1}{n}, \frac{i}{n}\right], \ i \in [n],$$

and $f_x(0) = x_1$. For $p \in [1, \infty)$, each $V_n$ is equipped with the normalized $\ell_p$-norms $\|x\|_{\overline{p}} = \left(\frac{1}{n}\sum_{i=1}^n |x_i|^p\right)^{1/p}$, which are compatible. Under the identification with step functions, this corresponds to the $L^p$ norm of functions, given by $\|f_x\|_p = \left(\int_0^1 |f_x(t)|^p dt\right)^{1/p}$. By Proposition A.3, this induces a norm on $V_\infty$. Consequently, the limit space can be identified with the $L^p$ space, specifically,

$$\overline{V_\infty} \cong L^p([0,1]) = \left\{f \colon [0,1] \to \mathbb{R} \text{ measurable} : \int_0^1 |f(t)|^p dt < \infty\right\}.$$

The permutations in $S_n$ act on functions on $L^p([0,1])$ by permuting consecutive intervals of length $1/n$. Formally, each $\sigma \in S_n$ defines a measure-preserving bijection $\widetilde{\sigma} \colon [0,1] \to [0,1]$ via $\widetilde{\sigma}((i-1)/n + t) = (\sigma(i)-1)/n + t$ for $t \in [0, 1/n)$ and $i \in [n]$ (and $\widetilde{\sigma}(1) = 1$ for simplicity).

A function $f \in \overline{V_\infty}$ gives rise to a probability measure $\mu_f$ defined as the distribution of $f(T)$ for $T$ uniformly sampled from $[0,1]$. Equivalently, $\mu_f = f_\# \lambda$ is the pushforward under $f$ of the Lebesgue measure $\lambda$ on $[0,1]$. All elements in the orbit of $f$ correspond to the same measure $\mu_f$, and conversely, two functions are in the orbit-closures of each other if and only if they correspond to the same measure. Thus, the orbit space $\overline{V_\infty}/G_\infty$ can be identified with $\mathcal{P}_p(\mathbb{R})$. Furthermore, the symmetrized distance coincides with the Wasserstein-$p$ distance in this case.

Likewise, for the $k$-fold direct sum of $\mathbb{V}_{\text{dup}}$ we fix an arbitrary norm on $\mathbb{R}^k$. The limit space $\overline{V_\infty}$ can be identified with $L^p([0,1]; \mathbb{R}^k)$, and the orbit space can be identified with $\mathcal{P}_p(\mathbb{R}^k)$ endowed with the Wasserstein $p$-distance with respect to $\|\cdot\|_{\mathbb{R}^k}$ in the same way. More details and proofs can be found in (Levin et al., 2025, Appx. F).

## C.2. Proof of Theorem 4.2

*Proof of Theorem 4.2.* We show that the model $\text{DeepSets}_\infty^{\rho;\sigma}$ can be decomposed as $\text{DeepSets}_\infty^{\rho;\sigma} = \sigma \circ \tau$ with $\tau \colon \overline{V_\infty}/G_\infty \to \mathbb{R}^r$ continuous. Since $\overline{V_\infty}/G_\infty$ is a metric space equipped with the symmetrized distance, continuity on $\overline{V_\infty}/G_\infty$ is equivalent to continuity on each of its compact subset (Willard, 2012). Finally, by definition, $\sigma \colon \mathbb{R}^r \to \mathbb{R}$ is a continuous function, this decomposition implies continuity of the entire model.

For any $n \in \mathbb{N}$, define the maps $\tau \colon V_\infty \to \mathbb{R}^r$ and $\tau_n \colon V_\infty \to \mathbb{R}^r$ by

$$\tau(X) = \sum_{i=1}^\infty \|X_{i:}\|_{\mathbb{R}^k}^p \rho(X_{i:}) \quad \text{and} \quad \tau_n(X) = \sum_{i=1}^{n-1} \|X_{i:}\|_{\mathbb{R}^k}^p \rho(X_{i:}),$$

respectively. We equip $\mathbb{R}^r$ with a standard norm $\|\cdot\|_{\mathbb{R}^r}$. Fix an arbitrary compact subset $K \subseteq \ell_p(\mathbb{R}^k)$. By Proposition 4.1, there exists $M > 0$ such that $\sup_{X \in K} \left(\sum_{j=1}^\infty \|X_{j:}\|_{\mathbb{R}^k}^p\right)^{1/p} \leq M$. This implies $\sup_{X \in K} \sup_{i \in \mathbb{N}} \|X_{i:}\|_{\mathbb{R}^k} \leq M$. Since $\rho \colon \mathbb{R}^k \to \mathbb{R}^r$ is continuous, it maps compact sets to compact sets; thus, there exists $M_\rho > 0$ such that $\sup_{X \in K} \sup_{i \in \mathbb{N}} \|\rho(X_{i:})\|_{\mathbb{R}^r} \leq M_\rho$. Each $\tau_n$ is continuous on $K$ as it is a finite sum of continuous functions (coordinate projections and $\rho$). According to the uniform tail decay property from Proposition 4.1, we have

$$\lim_{n \to \infty} \sup_{X \in K} \|\tau_n(X) - \tau(X)\|_{\mathbb{R}^r} \leq M_\rho \lim_{n \to \infty} \sup_{X \in K} \sum_{i \geq n} \|X_{i:}\|_{\mathbb{R}^k}^p = 0.$$

Hence, $\tau_n$ uniformly converges to $\tau$ on $K$. By invoking the uniform limit theorem (Rudin, 1976), we conclude that $\tau$ is continuous on $K$. Since $K$ was an arbitrary compact subset, we conclude that $\tau$ is continuous on all of $\overline{V_\infty}$. Furthermore, since $\tau$ is $G_\infty$-invariant, it induces a function on the orbit space $\overline{V_\infty}/G_\infty$, which we still denote by $\tau$. By Lemma A.4, this induced map $\tau \colon \overline{V_\infty}/G_\infty \to \mathbb{R}^r$ is continuous, which implies that $\text{DeepSets}_\infty^{\rho;\sigma} = \sigma \circ \tau$ is continuous. $\square$

## C.3. Proof of Theorem 4.3

To facilitate the proof, we introduce the set of invariant functions $\mathcal{F}_{\text{cont}}$ given by

$$\mathcal{F}_{\text{cont}} := \left\{ f : K/G_\infty \to \mathbb{R}, \ f(X) = \sigma\left(\sum_{i=1}^\infty \|X_{i:}\|_{\mathbb{R}^k}^p \rho(X_{i:})\right) \ \middle| \ r \in \mathbb{N}, \ \rho \in C\left(\mathbb{R}^k, \mathbb{R}^r\right), \ \sigma \in C\left(\mathbb{R}^r, \mathbb{R}\right) \right\}.$$

This class serves as an intermediate bridge between the target space $C(K/G_\infty)$ and the neural network class $\mathcal{F}_{\text{DS}}$.

The proof of Theorem 4.3 is split into two main components. We first invoke the Stone-Weierstrass Theorem to prove that $\mathcal{F}_{\text{cont}}$ is dense in $C(K/G_\infty)$. Next, we show that any element in $\mathcal{F}_{\text{cont}}$ can be uniformly approximated by $\mathcal{F}_{\text{DS}}$ through the universal approximation property of neural networks. Finally, the result follows by the transitivity of the density property. The details are organized into the three lemmas below, Lemma C.1, C.2, C.4.

**Lemma C.1.** *The function class $\mathcal{F}_{\text{cont}}$ is a subalgebra of $C(K/G_\infty)$.*

*Proof of Lemma C.1.* To establish that $\mathcal{F}_{\text{cont}}$ is a subalgebra of $C(K/G_\infty)$, we verify its closure under scalar multiplication, addition, and pointwise multiplication. Let $f_1, f_2 \in \mathcal{F}_{\text{cont}}$ with $f_j(X) = \sigma_j\left(\sum_{i=1}^\infty \|X_{i:}\|_{\mathbb{R}^k}^p \rho_j(X_{i:})\right)$ for $j \in \{1, 2\}$, where $\rho_j \in C(\mathbb{R}^k, \mathbb{R}^{r_j})$ and $\sigma_j \in C(\mathbb{R}^{r_j}, \mathbb{R})$.

**Scalar Multiplication.** For any $\lambda \in \mathbb{R}$, $(\lambda f_1)(X) = \sigma_\lambda\left(\sum_{i=1}^\infty \|X_{i:}\|_{\mathbb{R}^k}^p \rho_1(X_{i:})\right)$ where $\sigma_\lambda := \lambda \sigma_1$. Since $\sigma_\lambda$ is also continuous, $\lambda f_1 \in \mathcal{F}_{\text{cont}}$.

**Addition.** Define the concatenated map $\rho_0 : \mathbb{R}^k \to \mathbb{R}^{r_1 + r_2}$ as $\rho_0(x) = (\rho_1(x), \rho_2(x))$. Since its components $\rho_1$ and $\rho_2$ are continuous, $\rho_0 : \mathbb{R}^k \to \mathbb{R}^{r_1 + r_2}$ is continuous by the property of product topologies. Define $\sigma_{\text{add}} : \mathbb{R}^{r_1 + r_2} \to \mathbb{R}$ by $\sigma_{\text{add}}(u_1, u_2) = \sigma_1(u_1) + \sigma_2(u_2)$ for $u_1 \in \mathbb{R}^{r_1}, u_2 \in \mathbb{R}^{r_2}$. $\sigma_{\text{add}}$ is also continuous since the addition operator on $\mathbb{R}$ is continuous. Consequently,

$$(f_1 + f_2)(X) = \sigma_1\left(\sum_{i=1}^\infty \|X_{i:}\|_{\mathbb{R}^k}^p \rho_1(X_{i:})\right) + \sigma_2\left(\sum_{i=1}^\infty \|X_{i:}\|_{\mathbb{R}^k}^p \rho_2(X_{i:})\right) = \sigma_{\text{add}}\left(\sum_{i=1}^\infty \|X_{i:}\|_{\mathbb{R}^k}^p \rho_0(X_{i:})\right).$$

Thus, $\mathcal{F}_{\text{cont}}$ is closed under addition.

**Pointwise Multiplication.** Similarly, define the concatenated map $\rho_0 : \mathbb{R}^k \to \mathbb{R}^{r_1 + r_2}$ as $\rho_0(V) = (\rho_1(V), \rho_2(V))$, which is continuous. Define $\sigma_{\text{mul}} : \mathbb{R}^{r_1 + r_2} \to \mathbb{R}$ by $\sigma_{\text{mul}}(u_1, u_2) = \sigma_1(u_1) \cdot \sigma_2(u_2)$ for $u_1 \in \mathbb{R}^{r_1}, u_2 \in \mathbb{R}^{r_2}$, which is also continuous, because of the continuity of multiplication operator on $\mathbb{R}$. We then have

$$(f_1 \cdot f_2)(X) = \sigma_1\left(\sum_{i=1}^\infty \|X_{i:}\|_{\mathbb{R}^k}^p \rho_1(X_{i:})\right) \sigma_2\left(\sum_{i=1}^\infty \|X_{i:}\|_{\mathbb{R}^k}^p \rho_2(X_{i:})\right) = \sigma_{\text{mul}}\left(\sum_{i=1}^\infty \|X_{i:}\|_{\mathbb{R}^k}^p \rho_0(X_{i:})\right).$$

Thus, $\mathcal{F}_{\text{cont}}$ is closed under pointwise multiplication. $\square$

**Lemma C.2.** *$\mathcal{F}_{\text{cont}}$ separates points in $K/G_\infty$, and contains a nonzero constant function.*

To establish point separation on the quotient space $K/G_\infty$, we first characterize the conditions under which the metric $\overline{\text{d}}(X, Y)$ vanishes. Specifically, we identify the properties that distinguish points in distinct orbits. We claim that points in different orbits must differ in their first finite coordinate indices, see Claim C.3. Based on this, we can construct functions in $\mathcal{F}_{\text{cont}}$ to separate them.

**Claim C.3.** *For any compact $K \subseteq \ell_p(\mathbb{R}^k)$ and $X, Y \in K$, if the multisets $\{X_{i:} : \|X_{i:}\|_{\mathbb{R}^k} > \varepsilon\}$ and $\{Y_{i:} : \|Y_{i:}\|_{\mathbb{R}^k} > \varepsilon\}$ are equal for all $\varepsilon > 0$, then $\overline{\text{d}}(X, Y) = 0$.*

*Proof of Claim C.3.* Define the index sets by $I_x(\varepsilon) := \{i \in \mathbb{N} : \|X_{i:}\|_{\mathbb{R}^k} > \varepsilon\}$, and $I_y(\varepsilon) := \{i \in \mathbb{N} : \|Y_{i:}\|_{\mathbb{R}^k} > \varepsilon\}$, respectively. Since $X, Y \in K$, by Proposition 4.1, $\sup_{x \in K}\left(\sum_{i=1}^\infty \|X_{i:}\|_{\mathbb{R}^k}^p\right) < \infty$. Therefore, $I_x(\varepsilon)$ and $I_y(\varepsilon)$ are finite and have equal cardinality. Thus, there exists $\sigma_\varepsilon \in G_\infty$, such that $X_{\sigma_\varepsilon^{-1}(i):} = Y_{i:}$, for all $i \in I_y(\varepsilon)$. We then have

$$\overline{\text{d}}(X, Y)^p \leq \|\sigma_\varepsilon \cdot X - Y\|_p^p = \sum_{i \notin I_y(\varepsilon)} \|X_{\sigma_\varepsilon^{-1}(i):} - Y_{i:}\|_{\mathbb{R}^k}^p.$$

For $i \notin I_y(\varepsilon)$, we have $\|Y_{i:}\|_{\mathbb{R}^k} \leq \varepsilon$. Furthermore, since $\sigma_\varepsilon$ is a bijection, the index set $\{\sigma_\varepsilon^{-1}(i) : i \notin I_y(\varepsilon)\}$ is exactly $\{j \in \mathbb{N} : j \notin I_x(\varepsilon)\}$, where $\|X_{j:}\|_{\mathbb{R}^k} \leq \varepsilon$. Since $p \geq 1$, apply the inequality $\|a - b\|^p \leq 2^{p-1}(\|a\|^p + \|b\|^p)$ and obtain

$$\overline{\mathrm{d}}(X, Y)^p \leq 2^{p-1} \left( \sum_{i \notin I_x(\varepsilon)} \|X_{i:}\|_{\mathbb{R}^k}^p + \sum_{i \notin I_y(\varepsilon)} \|Y_{i:}\|_{\mathbb{R}^k}^p \right).$$

Letting $\varepsilon \to 0$ shows that $\overline{\mathrm{d}}(X, Y) = 0$, meaning $X$ and $Y$ represent the same point in $K/G_\infty$. $\qquad\square$

We are ready to prove separation of points.

*Proof of Lemma C.2.* First, $\mathcal{F}_{\mathrm{cont}}$ contains a nonzero constant function by taking $\sigma \equiv 1$, which yields $f(X) = 1$ for all $X$.

To establish point separation, consider two distinct points in $K/G_\infty$, represented by $X, Y \in K$ such that $\overline{\mathrm{d}}(X, Y) > 0$. By the contrapositive of Claim C.3, there exists $\varepsilon > 0$ such that the finite multisets $M_X := \{X_{i:} : \|X_{i:}\|_{\mathbb{R}^k} > \varepsilon\}$ and $M_Y := \{Y_{i:} : \|Y_{i:}\|_{\mathbb{R}^k} > \varepsilon\}$ are distinct. Thus, there exists some $v \in \mathbb{R}^k$ with $\|v\|_{\mathbb{R}^k} > \varepsilon$ such that their multiplicities differ: $m_X(v) \neq m_Y(v)$, where $m_X(v) := \#\{i \in \mathbb{N} : X_{i:} = v, X_{i:} \in M_X\}$ and $m_Y(v) := \#\{i \in \mathbb{N} : Y_{i:} = v, Y_{i:} \in M_Y\}$. Let $Q := (M_X \cup M_Y) \setminus \{v\}$, which is a finite set as both $M_X$ and $M_Y$ are finite. We define the separation radius

$$\eta := \min \left( \min_{u \in Q} \|u - v\|_{\mathbb{R}^k}, \|v\|_{\mathbb{R}^k} - \varepsilon \right).$$

Consider the ball $B(v, \eta)$ around $v$. For any $i \in \mathbb{N}$, we distinguish the following two cases.

**Case 1.** If $\|X_{i:}\|_{\mathbb{R}^k} \leq \varepsilon$, then by the reverse triangle inequality, $\|X_{i:} - v\|_{\mathbb{R}^k} \geq \|v\|_{\mathbb{R}^k} - \|X_{i:}\|_{\mathbb{R}^k} \geq \|v\|_{\mathbb{R}^k} - \varepsilon \geq \eta$.

**Case 2.** If $X_{i:} \in M_X$ and $X_{i:} \neq v$, then by the definition of $Q$, $\|X_{i:} - v\|_{\mathbb{R}^k} \geq \min_{u \in Q} \|u - v\|_{\mathbb{R}^k} \geq \eta$.

Combined, these cases imply that $X_{i:} \in B(v, \eta)$ if and only if $X_{i:} = v$. The same logic applies to the sequence $Y$. By Urysohn's Lemma (Willard, 2012), there exists a continuous function $\varphi \colon \mathbb{R}^k \to [0, 1]$, such that $\varphi\left(\overline{B(v, \frac{\eta}{2})}\right) = 1$ and $\varphi\left(\overline{\mathbb{R}^k \setminus B(v, \eta)}\right) = 0$. Define $f \in \mathcal{F}_{\mathrm{cont}}$ by setting $r = 1$, $\sigma(u) = u$, and $\rho = \varphi$, then,

$$f(X) = \sigma \left( \sum_{i=1}^\infty \|X_{i:}\|_{\mathbb{R}^k}^p \rho(X_{i:}) \right) = \|v\|_{\mathbb{R}^k}^p m_X(v) \neq \|v\|_{\mathbb{R}^k}^p m_Y(v) = \sigma \left( \sum_{i=1}^\infty \|Y_{i:}\|_{\mathbb{R}^k}^p \rho(Y_{i:}) \right) = f(Y).$$

Therefore, we conclude that $\mathcal{F}_{\mathrm{cont}}$ separates points in $K/G_\infty$. $\qquad\square$

**Lemma C.4.** $\mathcal{F}_{\mathrm{DS}}$ *is dense in* $\mathcal{F}_{\mathrm{cont}}$ *on* $K$.

*Proof of Lemma C.4.* We invoke the Universal Approximation Theorem (UAT) (Leshno et al., 1993): for any continuous non-polynomial activation function $\phi$, the set $\mathrm{NN}_{r,k}^\phi$ is dense in $C(\mathbb{R}^k, \mathbb{R}^r)$ under the supremum norm on compact sets. More formally, fix a compact set $Q \subseteq \mathbb{R}^k$ and a norm $\|\cdot\|_{\mathbb{R}^r}$ on $\mathbb{R}^r$. For any continuous function $h \in C(\mathbb{R}^k, \mathbb{R}^r)$, and any approximation precision $\varepsilon > 0$, there exists $\hat{h} \in \mathrm{NN}_{r,k}^\phi$ such that $\sup_{v \in Q} \|h(v) - \hat{h}(v)\|_{\mathbb{R}^r} \leq \varepsilon$.

Given any function $f \in \mathcal{F}_{\mathrm{cont}}$, it admits the form $f(X) = \sigma\left(\sum_{i=1}^\infty \|X_{i:}\|_{\mathbb{R}^k}^p \rho(X_{i:})\right)$, for some $r \in \mathbb{N}$, $\sigma \in C(\mathbb{R}^r, \mathbb{R})$, $\rho \in C(\mathbb{R}^k, \mathbb{R}^r)$. Since $K \in \ell_p(\mathbb{R}^k)$ is compact, by Proposition 4.1, there exists $M > 0$ such that $\sup_{X \in K} \left(\sum_{j=1}^\infty \|X_{j:}\|_{\mathbb{R}^k}^p\right)^{1/p} \leq M$. This implies $\sup_{X \in K} \sup_{i \in \mathbb{N}} \|X_{i:}\|_{\mathbb{R}^k} \leq M$. Since $\rho \in C(\mathbb{R}^k, \mathbb{R}^r)$, it maps compact sets to compact sets, i.e., there exists $M_\rho > 0$ such that $\sup_{X \in K} \sup_{i \in \mathbb{N}} \|\rho(X_{i:})\|_{\mathbb{R}^r} \leq M_\rho$. Thus for any $X \in K$, $\left\|\sum_{i=1}^\infty \|X_{i:}\|_{\mathbb{R}^k}^p \rho(X_{i:})\right\|_{\mathbb{R}^r} \leq M^p M_\rho$. This means that the input domains of functions $\sigma$ and $\rho$ are both compact, which we denote by $Q_\sigma$ and $Q_\rho$, respectively.

**Approximating $\sigma$.** For any $\varepsilon > 0$, UAT ensures there exists $\hat{\sigma} \in \mathrm{NN}_{1,r}^\phi$ such that $\sup_{u \in Q_\sigma} |\sigma(u) - \hat{\sigma}(u)| \leq \frac{\varepsilon}{2}$. Moreover, since $\hat{\sigma}$ is uniformly continuous on compact sets, there exists $\delta > 0$ such that $\|u_1 - u_2\|_{\mathbb{R}^r} \leq \delta \Rightarrow |\hat{\sigma}(u_1) - \hat{\sigma}(u_2)| \leq \frac{\varepsilon}{2}$.

**Approximating $\rho$.** Applying UAT to $\rho$, there exists $\hat{\rho} \in \text{NN}_{r,k}^{\phi}$ such that $\sup_{v \in Q_{\rho}} \|\rho(v) - \hat{\rho}(v)\|_{\mathbb{R}^r} \le \frac{\delta}{M^p}$. Consequently,

$$\sup_{X \in K} \left\| \sum_{i=1}^{\infty} \|X_{i:}\|_{\mathbb{R}^k}^p (\rho(X_{i:}) - \hat{\rho}(X_{i:})) \right\|_{\mathbb{R}^r} \le \sup_{X \in K} \sum_{i=1}^{\infty} \|X_{i:}\|_{\mathbb{R}^k}^p \sup_{X \in K} \|\rho(X_{i:}) - \hat{\rho}(X_{i:})\|_{\mathbb{R}^r} \le M^p \frac{\delta}{M^p} = \delta.$$

We define the overall approximating function $\hat{f} \in \mathcal{F}_{\text{DS}}$ by $\hat{f}(X) = \hat{\sigma}\left(\sum_{i=1}^{\infty} \|X_{i:}\|_{\mathbb{R}^k}^p \hat{\rho}(X_{i:})\right)$, then

$$
\begin{aligned}
\sup_{X \in K} \left| f(X) - \hat{f}(X) \right| &\le \sup_{X \in K} \left| \sigma\left(\sum_{i=1}^{\infty} \|X_{i:}\|_{\mathbb{R}^k}^p \rho(X_{i:})\right) - \hat{\sigma}\left(\sum_{i=1}^{\infty} \|X_{i:}\|_{\mathbb{R}^k}^p \rho(X_{i:})\right) \right| \\
&\quad + \left| \hat{\sigma}\left(\sum_{i=1}^{\infty} \|X_{i:}\|_{\mathbb{R}^k}^p \rho(X_{i:})\right) - \hat{\sigma}\left(\sum_{i=1}^{\infty} \|X_{i:}\|_{\mathbb{R}^k}^p \hat{\rho}(X_{i:})\right) \right| \\
&\le \frac{\varepsilon}{2} + \frac{\varepsilon}{2} = \varepsilon,
\end{aligned}
$$

which shows that $\mathcal{F}_{\text{DS}}$ uniformly approximates every function in $\mathcal{F}_{\text{cont}}$. By the definition of both of the function classes, $\mathcal{F}_{\text{DS}} \subseteq \mathcal{F}_{\text{cont}}$, therefore, $\mathcal{F}_{\text{DS}}$ is dense in $\mathcal{F}_{\text{cont}}$ on $K$ with respect to the supremum norm. $\qquad\square$

Finally, combining the above lemmas, we prove Theorem 4.3.

*Proof of Theorem 4.3.* Invoking the continuity property of our model, Theorem 4.2, $\mathcal{F}_{\text{cont}} \subseteq C(K/G_{\infty})$. By Proposition 3.1, the Stone–Weierstrass Theorem, and Lemmas C.1 and C.2, we conclude that $\mathcal{F}_{\text{cont}}$ is dense in $C(K/G_{\infty})$ with respect to the supremum norm. Combining this with Lemma C.4 and the transitivity of density, it follows that $\mathcal{F}_{\text{DS}}$ is dense in $C(K/G_{\infty})$, completing the proof that the model is universal. $\qquad\square$

### C.4. Proof of Theorem 4.5

*Proof of Theorem 4.5.* This theorem follows directly from the fact that $W_p$-convergence is equivalent to weak convergence in $\mathcal{P}_p$. Consider any sequence of measures $(\mu_i)_{i \in \mathbb{N}} \subseteq \mathcal{P}_p(\mathbb{R}^k)$ converging to $\mu$ in the Wasserstein $p$-metric, i.e., $W_p(\mu_i, \mu) \to 0$. According to the characterization of convergence in $\mathcal{P}_p(\mathbb{R}^k)$ (Villani et al., 2008), this is equivalent to $\int \varphi \, d\mu_i \to \int \varphi \, d\mu$ for all continuous functions $\varphi \colon \mathbb{R}^k \to \mathbb{R}$ satisfying the growth condition $|\varphi(x)| \le C(1 + \|x\|_{\mathbb{R}^k}^p)$ for some $C > 0$.

In our framework, $\rho \in \mathcal{F}_p(\mathbb{R}^k, \mathbb{R}^r)$ is a continuous mapping whose components $\rho_j$ satisfy $|\rho_j(x)| \le M_j(1 + \|x\|_{\mathbb{R}^k}^p)$. It follows that $\int \rho \, d\mu_i \to \int \rho \, d\mu$ as $i \to \infty$, establishing the continuity of the map $\mu \mapsto \int \rho \, d\mu$. Since $\sigma : \mathbb{R}^r \to \mathbb{R}$ is continuous, the composition $\overline{\text{DeepSets}}_{\infty}^{\rho,\sigma}(\mu) = \sigma(\int \rho \, d\mu)$ is continuous with respect to the $W_p$ topology. $\qquad\square$

### C.5. Proof of Theorem 4.6

We first define an intermediate class of functions, $\mathcal{F}_{\overline{\text{cont}}}$, to bridge $C(Q)$ and $\mathcal{F}_{\overline{\text{DS}}}$

$$\mathcal{F}_{\overline{\text{cont}}} := \left\{ f \colon Q \to \mathbb{R}, \ f(\mu) = \sigma\left(\int \rho \, d\mu\right) \ \Big| \ r \in \mathbb{N}, \ \rho \in \mathcal{F}_p\left(\mathbb{R}^k, \mathbb{R}^r\right), \ \sigma \in C\left(\mathbb{R}^r, \mathbb{R}\right) \right\}.$$

The proof strategy of Theorem 4.6 is similar to Theorem 4.3 by showing that: $(i)$ $\mathcal{F}_{\overline{\text{cont}}}$ is dense in $C(Q)$ via the Stone-Weierstrass Theorem, and $(ii)$ $\mathcal{F}_{\overline{\text{DS}}}$ is dense in $\mathcal{F}_{\overline{\text{cont}}}$ via the universal approximation property. We decompose the proof into three auxiliary Lemmas: C.5, C.6, and C.7.

**Lemma C.5.** *The function class $\mathcal{F}_{\overline{\text{cont}}}$ is a subalgebra of $C(Q)$.*

*Proof of Lemma C.5.* Invoking Theorem 4.5, $\mathcal{F}_{\overline{\text{cont}}} \subseteq C(Q)$. To show $\mathcal{F}_{\overline{\text{cont}}}$ is a subalgebra, it suffices to verify that it is closed under scalar multiplication, addition and pointwise multiplication. Let $f_1, f_2 \in \mathcal{F}_{\overline{\text{cont}}}$ with $f_j(\mu) = \sigma_j\left(\int \rho_j d\mu\right)$ for $j \in \{1,2\}$, where $\rho_j \in \mathcal{F}_p(\mathbb{R}^k, \mathbb{R}^{r_j})$ and $\sigma_j \in C(\mathbb{R}^{r_j}, \mathbb{R})$.

**Scalar Multiplication.** For any $\lambda \in \mathbb{R}$, $(\lambda f_1)(\mu) = \sigma_\lambda\left(\int \rho_1 d\mu\right)$ where $\sigma_\lambda := \lambda \sigma_1$. Since $\sigma_\lambda$ is also continuous, $\lambda f_1 \in \mathcal{F}_{\overline{\text{cont}}}$.

**Addition.** Define the concatenated map $\rho_0\colon \mathbb{R}^k \to \mathbb{R}^{r_1+r_2}$ as $\rho_0(x) = (\rho_1(x), \rho_2(x))$. Since its components $\rho_1$ and $\rho_2$ are continuous, $\rho_0\colon \mathbb{R}^k \to \mathbb{R}^{r_1+r_2}$ is continuous by the property of product topologies. Furthermore, since $\rho_j \in \mathcal{F}_p(\mathbb{R}^k, \mathbb{R}^{r_j})$ for $j \in \{1,2\}$, each individual component function of $\rho_0 = (\rho_{1,1}, \ldots, \rho_{1,r_1}, \rho_{2,1}, \ldots, \rho_{2,r_2})^\top$ satisfies the $p$-th order growth condition. Consequently, we have $\rho_0 \in \mathcal{F}_p(\mathbb{R}^k, \mathbb{R}^{r_1+r_2})$. Define $\sigma_{\text{add}}\colon \mathbb{R}^{r_1+r_2} \to \mathbb{R}$ by $\sigma_{\text{add}}(u_1, u_2) = \sigma_1(u_1) + \sigma_2(u_2)$ for $u_1 \in \mathbb{R}^{r_1}, u_2 \in \mathbb{R}^{r_2}$. Observe that $\sigma_{\text{add}}$ is also continuous since the addition operator on $\mathbb{R}$ is continuous. Consequently,

$$(f_1 + f_2)(\mu) = \sigma_1\left(\int \rho_1 d\mu\right) + \sigma_2\left(\int \rho_2 d\mu\right) = \sigma_{\text{add}}\left(\left[\int \rho_1 d\mu, \int \rho_2 d\mu\right]^\top\right) = \sigma_{\text{add}}\left(\int \rho_0 d\mu\right).$$

Thus, $\mathcal{F}_{\overline{\text{cont}}}$ is closed under addition.

**Pointwise Multiplication.** Similarly, define the concatenated map $\rho_0\colon \mathbb{R}^k \to \mathbb{R}^{r_1+r_2}$ as $\rho_0(V) = (\rho_1(V), \rho_2(V))$, which is continuous. Define $\sigma_{\text{mul}}\colon \mathbb{R}^{r_1+r_2} \to \mathbb{R}$ by $\sigma_{\text{mul}}(u_1, u_2) = \sigma_1(u_1) \times \sigma_2(u_2)$ for $u_1 \in \mathbb{R}^{r_1}, u_2 \in \mathbb{R}^{r_2}$, which is also continuous, because of the continuity of multiplication operator on $\mathbb{R}$. We then have

$$(f_1 \cdot f_2)(\mu) = \sigma_1\left(\int \rho_1 d\mu\right)\sigma_2\left(\int \rho_2 d\mu\right) = \sigma_{\text{mul}}\left(\left[\int \rho_1 d\mu, \int \rho_2 d\mu\right]^\top\right) = \sigma_{\text{mul}}\left(\int \rho_0 d\mu\right).$$

Thus, $\mathcal{F}_{\overline{\text{cont}}}$ is closed under pointwise multiplication. $\qquad\square$

**Lemma C.6.** *The function class $\mathcal{F}_{\overline{\text{cont}}}$ separates points in $Q$, and contains a nonzero constant function.*

*Proof of Lemma C.6.* The fact that $\mathcal{F}_{\overline{\text{cont}}}$ contains a nonzero constant function is trivial. To show it separates points in $Q$, suppose $\mu, \nu \in Q \subseteq \mathcal{P}_p(\mathbb{R}^k)$, and $\mu \neq \nu$. There must exist a Borel set $B \subseteq \mathbb{R}^k$ such that $\mu(B) \neq \nu(B)$. Without loss of generality, assume $\mu(B) > \nu(B)$, and let $\delta := \mu(B) - \nu(B) > 0$. Since $\mathbb{R}^k$ is a metric space, any measure in $\mathcal{P}_p(\mathbb{R}^k)$ is regular (Bogachev, 2007). By the property of regularity, for any $\varepsilon > 0$, there exists a closed set $E \subseteq B$ and an open set $O \supset B$ such that

$$\mu(B) \leq \mu(E) + \varepsilon, \quad \nu(B) \geq \nu(O) - \varepsilon.$$

Since $E$ and $\mathbb{R}^k \setminus O$ are disjoint closed sets in $\mathbb{R}^k$, by Urysohn's lemma (Willard, 2012) , there exists a continuous function $\varphi\colon \mathbb{R}^k \to [0,1]$ such that $\varphi(E) = 1$ and $\varphi(\mathbb{R}^k \setminus O) = 0$. It follows that

$$\int \varphi\, d\mu \geq \mu(E) \geq \mu(B) - \varepsilon, \quad \int \varphi\, d\nu \leq \nu(O) \leq \nu(B) + \varepsilon.$$

Set $\varepsilon = \frac{\delta}{3} > 0$, we obtain

$$\int \varphi\, d\mu - \int \varphi\, d\nu \geq \mu(B) - \nu(B) - 2\varepsilon = \frac{\delta}{3} > 0.$$

Recall that $\mathcal{F}_{\overline{\text{cont}}} := \left\{f\colon Q \to \mathbb{R}, \ f(\mu) = \sigma\left(\int \rho d\mu\right) \ \middle| \ r \in \mathbb{N}, \rho \in \mathcal{F}_p\left(\mathbb{R}^k, \mathbb{R}^r\right), \sigma \in C\left(\mathbb{R}^r, \mathbb{R}\right)\right\}$. By choosing $r = 1$, $\sigma(u) = u$, and $\rho = \varphi$, we observe that $\varphi$ is bounded and continuous, which implies $|\varphi(x)| \leq 1 \leq (1 + \|x\|_{\mathbb{R}^k}^p)$, and thus $\rho \in \mathcal{F}_p(\mathbb{R}^k, \mathbb{R})$. Therefore, it follows that for any $\mu, \nu \in Q \subseteq \mathcal{P}_p(\mathbb{R}^k)$ with $\mu \neq \nu$, there exists a function $f \in \mathcal{F}_{\overline{\text{cont}}}$ such that $f(\mu) \neq f(\nu)$. $\qquad\square$

**Lemma C.7.** *The function class $\mathcal{F}_{\overline{\text{DS}}}$ is dense in $\mathcal{F}_{\overline{\text{cont}}}$ on $Q$.*

*Proof of Lemma C.7.* The proof for this lemma proceeds by analogy with Lemma C.4 except for the following subtle issue: the compactness of $Q \subseteq \mathcal{P}_p(\mathbb{R}^k)$, as described in Proposition 4.4, does not ensure that the input domain of $\rho$ is compact. Consequently, the standard Universal Approximation Theorem cannot be directly applied. To address this issue, we leverage the property that compactness in $\mathcal{P}_p(\mathbb{R}^k)$ guarantees the probability mass in the tails of the measures is negligible. This allows us to replace $\rho$ with a function vanishing at infinity, which can then be uniformly approximated by neural networks on the entire space $\mathbb{R}^k$ using recent results for non-compact domains (van Nuland, 2024).

For any function $f \in \mathcal{F}_{\overline{\text{cont}}}$, we can represent it as $f(\mu) = \sigma\left(\int \rho d\mu\right)$, for some $r \in \mathbb{N}, \rho \in \mathcal{F}_p(\mathbb{R}^k, \mathbb{R}^r)$, and $\sigma \in C\left(\mathbb{R}^r, \mathbb{R}\right)$. We fix a compact set $Q \subseteq \mathcal{P}_p(\mathbb{R}^k)$, and let $\mathbb{R}^r$ be equipped with the Euclidean norm $\|\cdot\|_{\mathbb{R}^r}$. By the growth condition on $\rho$, there exist positive constants $M_j > 0$ such that for each component function $\rho_j$, we have $|\rho_j(x)| \leq M_j(1 + \|x\|_{\mathbb{R}^k}^p)$ for all $x \in \mathbb{R}^k$. Defining the vector $M := (M_1, \ldots, M_r)^\top \in \mathbb{R}^r$, it follows that $\|\rho(x)\|_{\mathbb{R}^r} \leq \|M\|_{\mathbb{R}^r}(1 + \|x\|_{\mathbb{R}^k}^p), \ \forall x \in \mathbb{R}^k$.

Moreover, Theorem 4.5 ensures the continuity of the map $\mu \mapsto \int \rho d\mu$. Thus, the image of the compact set $Q$ under this map, denoted by $Q_\sigma \subseteq \mathbb{R}^r$, is also compact, forming the input domain for $\sigma$.

**Approximating $\sigma$.** For any $\varepsilon > 0$, UAT ensures there exists $\hat{\sigma} \in \text{NN}_{1,r}^\phi$ such that $\sup_{u \in Q_\sigma} |\sigma(u) - \hat{\sigma}(u)| \le \frac{\varepsilon}{2}$. Moreover, since $\hat{\sigma}$ is uniformly continuous on compact sets, there exists $\delta > 0$ such that $\|u_1 - u_2\|_{\mathbb{R}^r} \le \delta \Rightarrow |\hat{\sigma}(u_1) - \hat{\sigma}(u_2)| \le \frac{\varepsilon}{2}$.

**Tail Control and Truncation of $\rho$.** Since $Q$ is compact, it is tight and $p$-uniformly integrable, by Proposition 4.4. Then for the constant $\frac{\delta}{4\|M\|_{\mathbb{R}^r}}$, there exists a positive constant $R > 0$ such that

$$\sup_{\mu \in Q} \mu\left(\mathbb{R}^k \setminus \overline{B(0, R)}\right) \le \frac{\delta}{4\|M\|_{\mathbb{R}^r}} \qquad \text{(tightness)},$$

$$\sup_{\mu \in Q} \int_{\mathbb{R}^k \setminus \overline{B(0,R)}} \|x\|_{\mathbb{R}^k}^p \, \mathrm{d}\mu \le \frac{\delta}{4\|M\|_{\mathbb{R}^r}} \qquad \text{($p$-uniformly integrability)}.$$

Using Urysohn's Lemma (Willard, 2012), there exists a continuous function $\varphi \colon \mathbb{R}^k \to [0, 1]$ such that $\varphi\left(\overline{B(0, R)}\right) = 1$ and $\varphi\left(\overline{\mathbb{R}^k \setminus B(0, 2R)}\right) = 0$. We truncate function $\rho$ with $\varphi$ by

$$\rho_\varphi(x) := \varphi(x)\rho(x), \quad \forall x \in \mathbb{R}^k.$$

Evidently, $\rho_\varphi \in C_0(\mathbb{R}^k, \mathbb{R}^r)$, the space of continuous functions vanishing at infinity. Furthermore, by the tail control of $\rho$, the discrepancy between $\rho$ and its truncated counterpart $\rho_\varphi$ can also be controlled on $Q$

$$\sup_{\mu \in Q} \int \|\rho_\varphi - \rho\|_{\mathbb{R}^r} d\mu \le \sup_{\mu \in Q} \int_{\mathbb{R}^k \setminus \overline{B(0,R)}} \|\rho(x)\|_{\mathbb{R}^r} d\mu(x) \le \|M\|_{\mathbb{R}^r} \sup_{\mu \in Q} \int_{\mathbb{R}^k \setminus \overline{B(0,R)}} \left(1 + \|x\|_{\mathbb{R}^k}^p\right) d\mu(x) \le \frac{\delta}{2}.$$

**Approximating $\rho_\varphi$.** For the approximation of $C_0$ functions, we invoke a specialized UAT (van Nuland, 2024): if the activation function $\phi$ is continuous, nonpolynomial, and asymptotically polynomial at $\pm\infty$, then any function in $C_0(\mathbb{R}^k, \mathbb{R})$ can be uniformly approximated by $\text{NN}_{1,k}^\phi$ on $\mathbb{R}^k$. This result extends naturally to the vector-valued space $C_0(\mathbb{R}^k, \mathbb{R}^r)$ and $\text{NN}_{r,k}^\phi$ by approximating each component function independently. Since $\rho_\varphi \in C_0\left(\mathbb{R}^k, \mathbb{R}^r\right)$, there exists $\hat{\rho} \in \text{NN}_{r,k}^\phi$ such that $\sup_{x \in \mathbb{R}^k} \|\hat{\rho}(x) - \rho_\varphi(x)\|_{\mathbb{R}^r} \le \frac{\delta}{2}$. Thus,

$$\sup_{\mu \in Q} \left\|\int (\rho - \hat{\rho}) d\mu\right\|_{\mathbb{R}^r} \le \sup_{\mu \in Q} \int \|\rho_\varphi - \rho\|_{\mathbb{R}^r} d\mu + \sup_{\mu \in Q} \int \|\rho_\varphi - \hat{\rho}\|_{\mathbb{R}^r} d\mu \le \delta.$$

We define the overall architecture as $\hat{f} = \hat{\sigma}(\int \hat{\rho} d\mu) \in \mathcal{F}_{\overline{\text{DS}}}$, and estimate the approximation error

$$\sup_{\mu \in Q} \left|f(\mu) - \hat{f}(\mu)\right| = \sup_{\mu \in Q} \left|\sigma\left(\int \rho d\mu\right) - \hat{\sigma}\left(\int \hat{\rho} d\mu\right)\right|$$

$$\le \sup_{\mu \in Q} \left|\sigma\left(\int \rho d\mu\right) - \hat{\sigma}\left(\int \rho d\mu\right)\right| + \sup_{\mu \in Q} \left|\hat{\sigma}\left(\int \rho d\mu\right) - \hat{\sigma}\left(\int \hat{\rho} d\mu\right)\right|$$

$$\le \frac{\varepsilon}{2} + \frac{\varepsilon}{2} = \varepsilon,$$

which concludes that $\mathcal{F}_{\overline{\text{DS}}}$ is dense in $\mathcal{F}_{\overline{\text{cont}}}$. $\qquad\qquad\square$

*Proof of Theorem 4.6.* By Proposition 3.1, the Stone–Weierstrass Theorem, and Lemmas C.5 and C.6, we conclude that $\mathcal{F}_{\overline{\text{cont}}}$ is dense in $C(Q)$. Combining this with Lemma C.7 and the transitivity of density, it follows that $\mathcal{F}_{\overline{\text{DS}}}$ is dense in $C(Q)$. $\qquad\qquad\square$

## D. Details and missing proofs from Section 4.2

In this section, we present additional details and proofs for our results on graphs.

### D.1. Duplication consistent sequence for graphs

We start by elaborating on the consistent sequence for graphs used in Section 4.2. More details and proofs for the following assertions can be found in (Levin et al., 2025, Appdx. G).

The duplication embedding consistent sequence for graphs $\mathbb{V}_{\mathrm{dup}}^G = \{(V_n), (\varphi_{N,n}), (\mathrm{G}_n)\}$ is defined as follows. The index set $(\mathbb{N}, \cdot \mid \cdot)$ is the set of natural numbers with divisibility partial order, where $n \preceq N$ if and only if $n \mid N$. For each $n \in \mathbb{N}$, $V_n = \mathbb{R}_{\mathrm{sym}}^{n \times n}$. For $n \preceq N$, the duplication embedding $\varphi_{N,n} \colon \mathbb{R}_{\mathrm{sym}}^{n \times n} \hookrightarrow \mathbb{R}_{\mathrm{sym}}^{N \times N}$ is given by $\varphi_{N,n}(A) = A \otimes \left(\mathbb{1}_{N/n}\mathbb{1}_{N/n}^{\top}\right)$, for $A \in \mathbb{R}_{\mathrm{sym}}^{n \times n}$, where $\otimes$ denotes the Kronecker product. The group is the symmetric group $S_n$ and the group embedding is the same as the case in Appendix C.1.2. $S_n$ acts on $V_n$ via $g \cdot A = gAg^{\top}$.

The space $V_\infty$ can be identified with the space of step graphons $W_A \colon [0,1]^2 \to \mathbb{R}$ given by

$$W_A(x,y) = A_{ij} \quad \text{for } (x,y) \in \left(\frac{i-1}{n}, \frac{i}{n}\right] \times \left(\frac{j-1}{n}, \frac{j}{n}\right], i,j \in [n].$$

The symmetric group acts on the induced step graphon by

$$\sigma \cdot W_A = W_A^{\sigma^{-1}} := W_A(\sigma^{-1}(x), \sigma^{-1}(y)).$$

We endow each $V_n$ with the cut norm, which is defined as

$$\|A\|_{\square} := \frac{1}{n^2} \max_{S \subseteq [n], T \subseteq [n]} \left| \sum_{i \in S, j \in T} A_{ij} \right| \quad \text{for } A \in \mathbb{R}_{\mathrm{sym}}^{n \times n}.$$

This cut norm extends to a norm in $V_\infty$, which coincides with the cut norm on graphons, defined as:

$$\|W\|_{\square} := \sup_{S,T \subseteq [0,1]} \left| \int_{S \times T} W(x,y) dx dy \right|.$$

The symmetrized distance coincides with the $\delta_{\square}$ distance, which is defined by

$$\overline{\mathrm{d}}(W,U) = \delta_{\square}(W,U) := \inf_{\varphi \in S_{[0,1]}} \|(U^\varphi - W)\|_{\square},$$

where $S_{[0,1]}$ denotes the set of all measure-preserving bijections $[0,1] \to [0,1]$; and $U^\varphi(x,y) := U(\varphi(x), \varphi(y))$. The orbit space can be identified with the space of symmetric measurable functions $[0,1]^2 \to \mathbb{R}$, modulo the equivalence relation $W_1 \sim W_2$ whenever $\delta_{\square}(W_1, W_2) = 0$.

### D.2. Proof of Theorem 4.9

*Proof of Theorem 4.9.* We expand the formulation (9) and get

$$h_m(W) = \int_{[0,1]^m} \prod_{1 \le i < j \le m} (a_{ij} W(x_i, x_j) + b_{ij}) \prod_{\ell=1}^m dx_\ell = \sum_{S \subseteq E_m} \left( \prod_{e \in S} a_e \prod_{e \notin S} b_e \right) t(S; W),$$

where $E_m$ denotes the set of edges of the $m$-node complete graph. Thus, each $h_m(\cdot)$ is a linear combination of homomorphism densities of simple graphs, which are continuous with respect to the cut distance $\delta_{\square}$ (Borgs et al., 2008, Thm. 2.7), and hence $\mathcal{F}_{\mathcal{W}} \subseteq C(\mathcal{W}_0, \delta_{\square})$ as claimed. $\square$

### D.3. Proof of Theorem 4.10

*Proof of Theorem 4.10.* To establish the density of $\mathcal{F}_{\mathcal{W}}$, it suffices to show that $\mathcal{HD} \subseteq \mathcal{F}_{\mathcal{W}}$, where $\mathcal{HD} := \mathrm{span}\{ t(F, \cdot) \mid F \text{ is a simple graph} \}$ is known to be dense in $C(\mathcal{W}_0, \delta_{\square})$ by (Diao et al., 2015, Thm. 2.2).

For any simple graph $F = (V, E)$ with $|V| = k \le m$, consider the functional $h_m(W)$ defined in (9) with parameters

$$a_{ij} = \begin{cases} 1 & \text{if } \{i,j\} \in E \\ 0 & \text{if } \{i,j\} \notin E \end{cases}, \quad \text{and} \quad b_{ij} = \begin{cases} 0 & \text{if } \{i,j\} \in E \\ 1 & \text{if } \{i,j\} \notin E \end{cases},$$

and note that $h_m(W) = t(F, W)$, proving that $\mathcal{HD} \subseteq \mathcal{F}_{\mathcal{W}}$ as desired. $\square$

## D.4. A universal deep model for graphs

First, we introduce a tensor contraction operator that serves as the nonlinear activation function in the network. For $k \in \mathbb{N}_0$, define a class of functions $\mathcal{G}_k := \{G : [0,1]^k \to \mathbb{R} \text{ measurable, bounded }\}/\sim_{\text{a.e.}}$ where $G_1 \sim_{\text{a.e.}} G_2$ if $G_1 = G_2$ almost everywhere, and set $\mathcal{G}_0 = \mathbb{R}$.

**Definition D.1.** *We define the operator* $T \colon \mathcal{G}_{k+1} \times \prod_{i=1}^{k} \mathcal{G}_2 \to \mathcal{G}_k$ *as*

$$(T(P, G_1, \ldots, G_k))(y_1, \ldots, y_k) = \int_{[0,1]} P(x, y_1, \ldots, y_k) \prod_{i=1}^{k} G_i(x, y_i) dx.$$

*For $k = 0$, the operator* $T \colon \mathcal{G}_1 \to \mathbb{R}$ *is given by* $\int_{[0,1]} P(x) dx$.

We define the $j$-th linear layer as

$$\mathcal{L}_j(P_j, W) := \begin{pmatrix} P_j \\ a_j^1 W + b_j^1 \\ \vdots \\ a_j^{k_j} W + b_j^{k_j} \end{pmatrix},$$

where $P_j \in \mathcal{G}_{k_j+1}$, for some $k_j \in \mathbb{N}_0$; $a_j^i, b_j^i \in \mathbb{R}$, for $i \in [k_j]$; $W \in \mathcal{W}_0$. We denote the $j$-th linear layer map as $\widetilde{\mathcal{L}}_j \colon \mathcal{G}_{k_j+1} \times \mathcal{W}_0 \to \left(\mathcal{G}_{k_j+1} \times \prod_{i=1}^{k_j} \mathcal{G}_2\right) \times \mathcal{W}_0$ given by $\widetilde{\mathcal{L}}_j(P_j, W) = (\mathcal{L}_j(P_j, W), W)$. Furthermore, we denote the nonlinearity in the network as $\widetilde{T} \colon \left(\mathcal{G}_{k+1} \times \prod_{i=1}^{k} \mathcal{G}_2\right) \times \mathcal{W}_0 \to \mathcal{G}_k \times \mathcal{W}_0$ given by $\widetilde{T}((P, G_1, \ldots, G_k), W) = (T(P, G_1, \ldots, G_k), W)$, where $T$ is the operator in Definition D.1.

We then contract the output of $\mathcal{L}_j$ using $T$ defined in Definition D.1 to get the input of $(j+1)$-th linear layer

$$T \circ \mathcal{L}_j(P_j, W) := P_{j+1} \in \mathcal{G}_{k_j}.$$

Suppose we have $m \geq 1$ layers, set $P_1 \in \mathcal{G}_m$, $P_1(x_1, \ldots, x_m) \equiv 1$, and let $k_j = m - j$. For the $j$-th tensor output, $1 \leq j \leq m+1$, we have $P_j \in \mathcal{G}_{m+1-j}$. The overall architecture $h_D^m : \mathcal{W}_0 \to \mathbb{R}$ can be written as

$$(h_D^m(W), W) = \widetilde{T} \circ \widetilde{\mathcal{L}}_m \circ \cdots \circ \widetilde{T} \circ \widetilde{\mathcal{L}}_1(P_1, W). \tag{15}$$

**Theorem D.2.** *Let $\mathcal{F}_{deep}$ be the class of functions defined as $\mathcal{F}_{deep} := \mathrm{span}\{h_D^m \mid m \in \mathbb{N}\}$, where each $h_D^m$ has the form of (15). Then $\mathcal{F}_{deep}$ is dense in $C(\mathcal{W}_0, \delta_\square)$.*

*Proof of Theorem D.2.* We establish the result by induction, demonstrating that the deep model form (15) recovers the structure of (9), and complete the proof by Theorem 4.10. We prove inductively that for any $r \in \mathbb{N}$ such that $2 \leq r \leq m+1$, the output tensor $P_r \in \mathcal{G}_{m+1-r}$ of the $r$th layer can be represented as

$$P_r(\mathbf{x}_{\geq r}) = \int_{[0,1]^{r-1}} \prod_{1 \leq i < j \leq r-1} L_i^{j-i}(W(x_i, x_j)) \prod_{i=1}^{r-1} \prod_{j=r}^{m} L_i^{j-i}(W(x_i, x_j)) d\mathbf{x}_{<r}, \tag{16}$$

where $L_i^v(W) := a_i^v W + b_i^v$ with $a_i^v, b_i^v \in \mathbb{R}$, $\mathbf{x}_{\geq r} := (x_r, \ldots, x_m)$, and $d\mathbf{x}_{<r} := dx_1 \ldots dx_{r-1}$. Having done so, we set $r = m+1$ and conclude that

$$P_{m+1} = \int_{[0,1]^m} \prod_{1 \leq i < j \leq m} L_i^{j-i}(W(x_i, x_j)) d\mathbf{x}_{<m+1} = \int_{[0,1]^m} \prod_{1 \leq i < j \leq m} \left(a_i^{j-i} W(x_i, x_j) + b_i^{j-i}\right) d\mathbf{x}_{<m+1}.$$

By setting $a_{ij} := a_i^{j-i}$ and $b_{ij} := b_i^{j-i}$, this expression exactly matches the form (9), completing the proof. Thus, we proceed to establish (16).

**Base Case ($r = 2$).** Directly applying Definition D.1, we have

$$P_2(x_2, \ldots, x_m) = \int_{[0,1]} \prod_{u=1}^{m-1} (a_1^u W(x_1, x_{u+1}) + b_1^u) \, dx_1 = \int_{[0,1]} \prod_{j=2}^{m} L_1^{j-1}(W(x_1, x_j)) dx_1$$

which is consistent with the inductive hypothesis (16).

**Inductive Step.** Suppose the hypothesis holds for some $r \in \mathbb{N}$ ($2 \leq r \leq m$), such that $P_r \in \mathcal{G}_{m+1-r}$ takes the form of (16). Then, for $r + 1$, we obtain

$$P_{r+1}(x_{r+1}, \ldots, x_m)$$

$$= \int_{[0,1]} P_r(x_r, x_{r+1}, \ldots, x_m) \prod_{u=1}^{m-r} (a_r^u W(x_r, x_{r+u}) + b_r^u) \, dx_r$$

$$= \int_{[0,1]} \left( \int_{[0,1]^{r-1}} \prod_{1 \leq i < j \leq r-1} L_i^{j-i}(W(x_i, x_j)) \prod_{i=1}^{r-1} \prod_{j=r}^{m} L_i^{j-i}(W(x_i, x_j)) d\mathbf{x}_{<r} \right) \prod_{j=r+1}^{m} L_r^{j-r}(W(x_r, x_j)) dx_r$$

$$= \int_{[0,1]^r} \left( \prod_{1 \leq i < j \leq r-1} L_i^{j-i}(W(x_i, x_j)) \right) \left( \prod_{i=1}^{r-1} L_i^{r-i}(W(x_i, x_r)) \right) \left( \prod_{i=1}^{r} \prod_{j=r+1}^{m} L_i^{j-i}(W(x_i, x_j)) \right) d\mathbf{x}_{<r+1}$$

$$= \int_{[0,1]^r} \prod_{1 \leq i < j \leq r} L_i^{j-i}(W(x_i, x_j)) \prod_{i=1}^{r} \prod_{j=r+1}^{m} L_i^{j-i}(W(x_i, x_j)) d\mathbf{x}_{<r+1}.$$

This confirms that the hypothesis holds for $r + 1$. □

We remark that the above architecture can be made more expressive by allowing general widths for the linear layers, and by using general linear equivariant maps (Herbst & Jegelka, 2025). The latter would require adapting the nonlinear operators in Definition D.1 appropriately, and we do not further explore these extensions.

## E. Details and missing proofs from Section 4.3

### E.1. Duplication consistent sequence for point clouds

We elaborate on the consistent sequence used in Section 4.3. See (Levin et al., 2025, Appx. H) for more details and proofs of the following assertions.

Similar to the case of graphs, the duplication embedding consistent sequence for point clouds $\mathbb{V}_{\text{dup}}^P = \{(V_n), (\varphi_{N,n}), (G_n)\}$ is defined as follows. The index set $(\mathbb{N}, \cdot \mid \cdot)$ is the set of natural numbers with divisibility partial order, where $n \preceq N$ if and only if $n \mid N$. For each $n \in \mathbb{N}$, $V_n = \mathbb{R}^{n \times k}$, which represents sets of $n$ points in $\mathbb{R}^k$, and $k$ is fixed. The group is $G_n = S_n \times O(k)$, where $S_n$ is the permutation group, and $O(k)$ is the orthogonal group. The group action on $V_n$ is defined as

$$(g, h) \cdot X = gXh^\top.$$

For $n \preceq N$, the duplication embedding $\varphi_{N,n} \colon \mathbb{R}^{n \times k} \hookrightarrow \mathbb{R}^{N \times k}$ is given by $\varphi_{N,n}(X) = X \otimes \mathbb{1}_{N/n}$, and the group embedding $\theta_{N,n} \colon S_n \times O(k) \hookrightarrow S_N \times O(k)$ is given by $\theta_{N,n}(g, h) = (g \otimes I_{N/n}, h)$. We consider the Euclidean norm on $\mathbb{R}^k$ denoted by $\| \cdot \|_{\mathbb{R}^k}$, which corresponds to the inner product preserved by elements of $O(k)$. We equip each $V_n$ with the normalized $\ell_2$ norm:

$$\|X\|_{\bar{2}} = \left( \frac{1}{n} \sum_{i=1}^{n} \|X_{i:}\|_{\mathbb{R}^k}^2 \right)^{1/2}.$$

Similarly to the case of sets, we can identify each matrix $X \in \mathbb{R}^{n \times k}$ with a step function $[0, 1] \to \mathbb{R}^k$. Then the limit space can be identified with $\overline{V_\infty} = L^2\left([0, 1]; \mathbb{R}^k\right)$, and the symmetrized metric can be written as:

$$\bar{d}(X, Y) = \inf_{h \in O(k)} \inf_{\varphi \in S_{[0,1]}} \left( \int_0^1 \|h(X(t)) - Y(\varphi(t))\|_{\mathbb{R}^k}^2 dt \right)^{1/2},$$

where $S_{[0,1]}$ is the set of measure preserving bijections.

We can further identify each matrix $X \in \mathbb{R}^{n \times k}$ with an empirical probability measure in $\mathbb{R}^k$. The space of orbit closures $\overline{V_\infty}/G_\infty$ can then be identified with the space of orbits of probability measures on $\mathbb{R}^k$ under the $O(k)$ action by pushforwards $h \cdot \mu = h_\# \mu$ for $h \in O(k)$. From this perspective, we can further rewrite the symmetrized distance as

$$\bar{d}(X, Y) = \inf_{h \in O(k)} W_2(h \cdot \mu_X, \mu_Y) \quad \text{for } X, Y \in \overline{V_\infty},$$

where $\mu_X = X_{\#}\lambda$, $\lambda$ is the Lebesgue measure on $[0,1]$.

## E.2. Proof of Proposition 4.11

*Proof of Proposition 4.11.* Consider the set of probability measures corresponding to the set $K_R$, denoted by $Q_R :=$ $\{X_{\#}\lambda \mid X \in K_R\}$. We show that $Q_R$ is equal to the set of probability measures with $R$-bounded support $\mathcal{P}_2\left(\overline{B(0,R)}\right) :=$ $\{\mu \in \mathcal{P}_2\left(\mathbb{R}^k\right) \mid \operatorname{supp}\mu \subseteq \overline{B(0,R)}\}$.

First, we show that $Q_R \subseteq \mathcal{P}_2\left(\overline{B(0,R)}\right)$. For any $\mu \in Q_R$, there exists a function $X \in K_R$ such that $\mu = X_{\#}\lambda$. By the definition of the push-forward measure, for any Borel set $B \subseteq \mathbb{R}^k$, we have $\mu(B) = \lambda(X^{-1}(B))$. Since $X \in K_R$, for almost every $t \in [0,1]$ we have $\|X(t)\|_{\mathbb{R}^k} \leq R$. Consequently, $\operatorname{supp}\mu \subseteq \overline{B(0,R)}$, which implies $Q_R \subseteq \mathcal{P}_2\left(\overline{B(0,R)}\right)$.

Conversely, we show that $\mathcal{P}_2\left(\overline{B(0,R)}\right) \subseteq Q_R$. For any $\mu \in \mathcal{P}_2\left(\overline{B(0,R)}\right)$, there exists a Borel mapping $X : [0,1] \to \overline{B(0,R)}$ such that $X_{\#}\lambda = \mu$ because $\overline{B(0,R)}$ is a standard Borel space (Srivastava, 1998, Thm. 3.3.13). Since the image of $X$ is contained within the ball, we have $\|X(t)\|_{\mathbb{R}^k} \leq R$ for all $t$. Moreover, since $\int_{[0,1]}\|X(t)\|_{\mathbb{R}^k}^2 d\lambda(t) \leq R^2 < \infty$, it follows that $X \in L^2([0,1];\mathbb{R}^k)$, and thus $X \in K_R$. This concludes that $\mathcal{P}_2\left(\overline{B(0,R)}\right) \subseteq Q_R$.

Consequently, the set $Q_R = \mathcal{P}_2\left(\overline{B(0,R)}\right)$ is a compact subset of the Wasserstein space $\mathcal{P}_2\left(\mathbb{R}^k\right)$ with respect to the $W_2$ metric by Proposition 4.4. Since the canonical projection map $\pi : \mathcal{P}_2\left(\mathbb{R}^k\right) \to \mathcal{P}_2\left(\mathbb{R}^k\right)/\mathrm{O}(k)$ is continuous (see the proof of Lemma A.4), we conclude that $K_R/G_\infty$, which can be identified with $Q_R/\mathrm{O}(k)$, is compact in $\overline{V_\infty}/G_\infty$. $\qquad\square$

## E.3. Proof of Theorem 4.12

*Proof of Theorem 4.12 .* By (Herbst & Jegelka, 2025), the IGN architecture in (13) is continuous with respect to $\delta_2$ distance. Therefore, to establish the overall model is continuous, it suffices to show that $X \mapsto W_X$ is continuous with symmetrized distance $\overline{\mathrm{d}}$ in its input and $\delta_2$ distance in its image. For $X, Y \in K_R = \left\{X \in L^2\left([0,1];\mathbb{R}^k\right) : \|X\|_\infty \leq R\right\}$, the distance of their corresponding covariance functions can be bounded by

$$\delta_2(W_X, W_Y)^2 = \frac{1}{4R^4}\inf_{\varphi \in S_{[0,1]}}\left(\int_{[0,1]^2}|\langle X(x), X(y)\rangle - \langle Y(\varphi(x)), Y(\varphi(y))\rangle|^2 dxdy\right)$$

$$= \frac{1}{4R^4}\inf_{\varphi \in S_{[0,1]}}\left(\int_{[0,1]^2}|\langle X(x) - Y(\varphi(x)), X(y)\rangle + \langle Y(\varphi(x)), X(y) - Y(\varphi(y))\rangle|^2 dxdy\right)$$

$$\leq \frac{1}{4R^2}\inf_{\varphi \in S_{[0,1]}}\left(\int_{[0,1]^2}(\|X(x) - Y(\varphi(x))\|_{\mathbb{R}^k} + \|X(y) - Y(\varphi(y))\|_{\mathbb{R}^k})^2 dxdy\right).$$

Applying Young's inequality, namely, $(a+b)^2 \leq 2(a^2 + b^2)$, we obtain

$$\delta_2(W_X, W_Y)^2 \leq \frac{1}{2R^2}\inf_{\varphi \in S_{[0,1]}}\left(\int_{[0,1]^2}\left(\|X(x) - Y(\varphi(x))\|_{\mathbb{R}^k}^2 + \|X(y) - Y(\varphi(y))\|_{\mathbb{R}^k}^2\right)dxdy\right)$$

$$= \frac{1}{R^2}\inf_{\varphi \in S_{[0,1]}}\left(\int_0^1\|X(x) - Y(\varphi(x))\|_{\mathbb{R}^k}^2 dx\right).$$

For any $h \in \mathrm{O}(k)$, we have $W_{h\cdot X} = \langle h\cdot X(x), h\cdot X(y)\rangle = \langle X(x), X(y)\rangle = W_X$, so

$$\delta_2(W_X, W_Y)^2 = \delta_2(W_{h\cdot X}, W_Y)^2 \leq \frac{1}{R^2}\inf_{\varphi \in S_{[0,1]}}\left(\int_0^1\|h\cdot X(x) - Y(\varphi(x))\|_{\mathbb{R}^k}^2 dx\right).$$

Taking the infimum over the orthogonal group $\mathrm{O}(k)$, the inequality still holds:

$$\delta_2(W_X, W_Y) \leq \frac{1}{R}\inf_{h \in \mathrm{O}(k)}\inf_{\varphi \in S_{[0,1]}}\left(\int_0^1\|h\cdot X(x) - Y(\varphi(x))\|_{\mathbb{R}^k}^2 dx\right)^{1/2} = \frac{1}{R}\overline{\mathrm{d}}(X,Y),$$

which implies that the map $X \mapsto W_X$ is Lipschitz continuous. $\qquad\square$

**E.4. Proof of Theorem 4.13**

The proof relies on the correspondence between $K_R/G_\infty$ and the orbit space of $(\mathcal{W}_0, \delta_2)$ under the action of the orthogonal group $O(k)$, together with the universality of the model $I^\varrho_{\varrho,M,L,b}$ on compact subsets of $(\mathcal{W}_0, \delta_2)$ (Herbst & Jegelka, 2025). We first state and prove several auxiliary lemmas.

**Lemma E.1.** *For $X, Y \in K_R$, and $W_X, W_Y$ are the corresponding covariance functions defined in (12), then $\overline{\mathrm{d}}(X, Y) = 0$ if and only if $\delta_2(W_X, W_Y) = 0$.*

This lemma not only ensures that the architecture is well-defined, but also implies that point separation on $K_R/G_\infty$ is equivalent to separate points on $(\mathcal{W}_0, \delta_2)/O(k)$. The proof is built on the following claim that if the covariance functions are equal almost everywhere, then there must exist an orthogonal transformation between the original functions. More formally,

**Claim E.2.** *Let $X, Y \in L^2([0,1], \mathbb{R}^k)$. Suppose that*

$$\langle X(x), X(y) \rangle = \langle Y(x), Y(y) \rangle \quad \text{for almost every } x, y \in [0,1].$$

*Then there exists an orthogonal transformation $h \in O(k)$ such that $Y = hX$ almost everywhere.*

*Proof of Claim E.2.* First, we can identify the space $L^2([0,1], \mathbb{R}^k)$ with $\mathcal{B}(L^2([0,1]), \mathbb{R}^k)$, which is the set of bounded linear operators between two Hilbert spaces $L^2([0,1]) \to \mathbb{R}^k$, equipped with the Hilbert–Schmidt norm $\| \cdot \|_{HS}$. In more detail, each $X \in L^2([0,1], \mathbb{R}^k)$ can be written as

$$X = (X_1, \ldots, X_k)^\top \quad \text{with} \quad X_i \in L^2([0,1]),$$

which defines a bounded linear map $\Phi \colon L^2([0,1]) \to \mathbb{R}^k$ by

$$\Phi_X \varphi = (\langle X_1, \varphi \rangle, \ldots, \langle X_k, \varphi \rangle)^\top.$$

Conversely, by Riesz Representation Theorem any bounded linear operator $\Phi \in \mathcal{B}(L^2([0,1]), \mathbb{R}^k)$ can be written as this form for some $X_1, \cdots, X_k \in L^2([0,1])$.

Define $\mathcal{V}_m := \mathrm{span}\{X_1, \ldots, X_k\}$. Then $\Phi_X$ vanishes on the orthogonal complement $\mathcal{V}_k^\perp$. Since $\Phi_X \colon \mathcal{V}_k \to \mathbb{R}^k$ is a linear map between finite-dimensional vector spaces, it admits a singular value decomposition (Friedberg et al., 1997). Thus, there exist non-negative singular values $\sigma_1, \ldots, \sigma_k \in \mathbb{R}_{\geq 0}$, orthonormal bases $\{v_1, \ldots, v_k\} \subseteq \mathbb{R}^k$, and $\{u_1, \ldots, u_k\} \subseteq L^2([0,1])$ such that

$$\Phi_X = \sum_{i=1}^k \sigma_i \langle u_i, \cdot \rangle v_i.$$

Moreover, for each $\sigma_i > 0$, the $u_i$ is an eigenvector of the self-adjoint operator $\Phi_X^* \Phi_X$ corresponding to the eigenvalue $\sigma_i^2$. For indices $i$ with $\sigma_i = 0$, the vectors $u_i$ can be chosen to complete the set $\{u_1, \ldots, u_k\}$ into an orthonormal basis of $\mathcal{V}_k$.

The self-adjoint operator $\Phi_X^* \Phi_X$ evaluated on $\varphi \in L^2([0,1])$ yields

$$(\Phi_X^* \Phi_X \varphi)(x) = \sum_{i=1}^k \langle X_i, \varphi \rangle X_i(x) = \sum_{i=1}^k X_i(x) \int_0^1 X_i(y)\varphi(y)dy = \int_0^1 \langle X(x), X(y) \rangle \varphi(y)dy.$$

Since $\langle X(x), X(y) \rangle = \langle Y(x), Y(y) \rangle$ almost everywhere, then $\Phi_X^* \Phi_X = \Phi_Y^* \Phi_Y$. Therefore, we can choose the same orthonormal bases $\{u_1, \ldots, u_k\} \subseteq L^2([0,1])$ for $\Phi_X$ and $\Phi_Y$, such that

$$\Phi_X = \sum_{i=1}^k \sigma_i \langle u_i, \cdot \rangle v_i, \quad \Phi_Y = \sum_{i=1}^k \sigma_i \langle u_i, \cdot \rangle w_i,$$

where $\{v_1, \ldots, v_k\}$ and $\{w_1, \ldots, w_k\}$ are both the orthonormal bases in $\mathbb{R}^k$. Then there exists $h \in O(k)$ such that $h(v_i) = w_i$, which implies

$$h\Phi_X = \sum_{i=1}^k \sigma_i \langle u_i, \cdot \rangle (hv_i) = \sum_{i=1}^k \sigma_i \langle u_i, \cdot \rangle w_i = \Phi_Y.$$

For any $\omega \in \mathbb{R}^k$, $\langle Y(\cdot), \omega \rangle = \Phi_Y^* \omega = (h\Phi_X)^* \omega = \langle hX(\cdot), \omega \rangle$, which yields $Y = hX$ in $L^2([0,1], \mathbb{R}^k)$. $\qquad \square$

*Proof of Lemma E.1.* First, by Theorem 4.12, if $\overline{\mathrm{d}}(X, Y) = 0$, then $\delta_2(W_X, W_Y) = 0$. We only need to show that

$\delta_2(W_X, W_Y) = 0$ implies $\overline{d}(X, Y) = 0$. Since $\delta_2(W_X, W_Y) = 0$, there exists measure-preserving maps $\varphi, \psi \in \bar{S}_{[0,1]}$, such that $W_X^\varphi = W_Y^\psi$ almost everywhere (Lovász, 2012, Cor. 10.35), hence

$$\langle X(\varphi(x)), X(\varphi(y)) \rangle = \langle Y(\psi(x)), Y(\psi(y)) \rangle \quad \text{for almost every } x, y \in [0, 1].$$

By Claim E.2, there exists $h \in O(k)$ such that $hX^\varphi = Y^\psi$, where $X^\varphi(x) = X(\varphi(x))$. Therefore,

$$\overline{d}(X, Y) = \inf_{h \in O(k)} \inf_{\varphi \in S_{[0,1]}} \left( \int_0^1 \|h(X(t)) - Y(\varphi(t))\|_{\mathbb{R}^k}^2 dt \right)^{1/2}$$

$$= \inf_{h \in O(k)} \inf_{\varphi, \psi \in \overline{S}_{[0,1]}} \left( \int_0^1 \|h(X(\varphi(t))) - Y(\psi(t))\|_{\mathbb{R}^k}^2 dt \right)^{1/2}$$

$$\leq 0,$$

where $\overline{S}_{[0,1]}$ denotes the set of measure-preserving maps $[0, 1] \to [0, 1]$. This proves that $\overline{d}(X, Y) = 0$, as desired. $\qquad\square$

*Proof of Theorem 4.13.* We define the class of functions $\mathcal{F}_h$ as

$$\mathcal{F}_h := \text{span}\{X \mapsto t(F, W_X) \mid F \text{ is simple }\},$$

where $t$ denotes the homomorphism density defined in (8). It is straightforward to see that $\mathcal{F}_h$ is a subalgebra, since $t(F_1, \cdot)\, t(F_2, \cdot) = t(F_1 \sqcup F_2, \cdot)$. By Lemmas E.1 and (Lovász, 2012, Cor. 10.34), $\mathcal{F}_h$ separates points. Therefore, $\mathcal{F}_h$ is dense in $C(K_R/G_\infty)$.

By Theorem 4.12, $X \mapsto W_X$ is a continuous mapping. Since the input $K_R/G_\infty$ is compact by Proposition 4.11, the image of $K_R/G_\infty$ is a compact subset of $(\mathcal{W}_0, \delta_2)$. The IGN model is universal on compact subsets of $(\mathcal{W}_0, \delta_2)$, and thus can approximate arbitrarily well any homomorphism density of simple graphs (Herbst & Jegelka, 2025). Consequently, $\mathcal{F}_{PC}^\theta$ is dense in $C(K_R/G_\infty)$. $\qquad\square$

