# OpenReview forum: "Any-dimensional invariant universality"
_ICML.cc/2026/Conference — ICML 2026 regular_

### Official Review · Reviewer_BxZS · 2026-02-25

**Soundness:** 4
**Presentation:** 3
**Significance:** 4
**Originality:** 4
**Overall Recommendation:** 6
**Confidence:** 5

**Summary:**

The very classical and celebrated universal approximation theorem states that any sufficiently nice function can be approximated to arbitrary precision by a neural network. Implicitly here, our function is defined on some fixed and finite dimensional vector space. Often times, however, one is faced with situations where they must work with architectures for which the inputs can come from vector spaces of varying dimension. It is then natural to ask whether the universal approximation theorem can be generalized to this context. That is to say, given any any-dimensional architecture, can it be made to approximate a sequence of functions on the constituent spaces to arbitrary precision.

On the one had intuition tells us that this should not be possible. After all, if a model is trained on only finite amounts of data, how can it have any hope of generalizing behaviors in dimensions it hasn't seen. In the following manuscript, the authors provide a mathematical framework for performing any dimensional approximation.

**Compliance With Llm Reviewing Policy:**

Affirmed.

**Final Justification:**

My score remains unchanged through the discussion period.

On the positive side, The mathematics of this work is quite sophisticated. It also finds itself quite firmly rooted in a new research program that aims to contend with the any-dimensional input problem. The presentation is more or less clear, in that every claim naturally follows what proceeds it. The appendices are detailed and robust leading to a presentation that feels complete and meaningful

On the negative side, I did find that the authors would sometimes get lost in the formality of their machinery. Even as someone who is familiar with the any-dimensional input problem and this approach through functor algebras, I definitely found myself needing to pause to make sure I still understood why things were happening. I also saw that other reviewers raised concerns about practical implementation of the methods. I personally value the theory enough that this was less of an issue for me.

**Key Questions For Authors:**

This is a beautiful paper, but there were a few very small things that I would hope the authors could remark upon (or fix).

1.) I believe this is a typo, but on the top of page 3 you state, "Let $P(\mathbb{R}^n)$ be the set of probability measures on $\mathcal{X}$" should one of these either be an $\mathcal{X}$ or a $\mathbb{R}^n$?

2.) I have a few questions about your definition of consistent sequence. Firstly, everything is defined in terms of a choice of directed poset structure on $\mathbb{N}$. Why is this the case? Certainly there are many other directed posets that one might choose to consider sequences over, and it is unclear to me what property of $\mathbb{N}$ specifically you are using. Secondly, the definition of "directed poset" that you give in the footnote is incorrect, or at least worded in a way that I found strange. It should read either that every two elements have a \emph{common} upper bound, or otherwise that every \emph{pair} of elements has an upper bound.

3.) Keeping with the second point, it feels like your notion of a consistent sequence relates very closely to the mathematical notion of persistence (i.e. functors from poset categories to categories of vector spaces). Can this relationship be made more precise and perhaps even leveraged?

**Limitations:**

Yes

**Strengths And Weaknesses:**

Soundness: The mathematical content of this work is perfectly sound from what I can tell. This content is deeply theoretical, pulling from a very wide range of mathematical disciplines. Proofs for all statements are either provided, or given references for. My only minor complaint is the lack of empirical data that shows the theoretical results of this work in action. Of course, this is ultimately a theory paper, so this isn't a critical flaw, but it would have been nice to see some data.

Presentation: The work is presented in what I feel is an appropriately formal and rigorous way. The authors take the time to motivate most of what is being done, and often times break their exposition in smaller more digestible chunks. I will admit, however, that there were times where the mathematical formalism became a bit much. I am very familiar with the material, so this never made it impossible for me to read the work, but I found myself repeatedly thinking that someone coming to this material for the first time might have a tough time getting through it. Considering how inherently technical the theory underlying this work is, I understand it is absolutely not an easy task to make it more approachable (especially given the space constraints).

Significance: The "any dimensional inputs" problem is one of the most significant in theoretical machine learning. The present work -- which follows in the footsteps of a very recent research program started by Levin, Diaz, and others --  is in my mind likely to be truly ground breaking. The way that the authors weave so many disparate threads in mathematics to produce their theory is truly revolutionary at times. The authors repeatedly note that many classical constructions in any-dimensional input architectures do not generalize to their setting, and I find this to be a feature not a bug. As stated by the authors, these classical constructions are only natural in so far as one thinks exclusively about individual vector spaces. To truly approximate an any-dimensional input, one must shift their perspective away from the individual spaces and toward a more holistic object that encapsulates all of them. This change in perspective, which is the core technical underpinning of this work, has to potential to be monumentally influential to those who consider these problems.

Originality: As stated above, the way that the authors have combined so many different mathematical fields is incredible. The only thing that slightly takes away from the "originality" score is that this work sits firmly in a research program that was only started in the last few years.

---

> ### Author Rebuttal · Authors · 2026-03-30
>
> We sincerely thank the reviewer for the detailed and enthusiastic review.
>
> **W1 (Lack of empirical data).**
> Thank you for raising this concern! As we explain in more detail in our response to Reviewer VBA5, the architectures we consider in this paper are mostly small modifications of widely used and studied architectures from the literature. Their empirical performance is already well-demonstrated, and our paper instead studies their universality properties. Universality, in turn, is an important but fundamentally theoretical and asymptotic property, which is therefore intractable to demonstrate or test numerically.
>
> **W2 (Typo: probability measures).**
> Indeed, this was a typo, thank you for catching it! We meant to only consider probability measures on $\mathbb{R}^n$ and will revise this in the next version.
>
> **Q1 (Why directed poset on N?).**
> This is a very interesting question! Indeed, all the definitions and our general recipe for proving universality go through for consistent sequences indexed by general posets (and, incidentally, so does the framework of [1] on which we are relying). On the other hand, all the consistent sequences we are aware of consist of countably-many vector spaces, and can be naturally indexed by a poset structure on $\mathbb{N}$. In fact, all our examples involve just two such poset structures, the usual one and the divisibility partial order. Since the mathematical formalism gets somewhat involved as you observed, we opted to keep the technical baggage to a necessary minimum and only discuss posets on $\mathbb{N}$.
>
> It would be extremely interesting to find naturally-occurring consistent sequences indexed by other posets, as studying their properties would help crystallize the properties of the poset that are really needed for any-dimensionality.
>
> **Q1 cont. (Definition of directed poset).**
> Indeed, thank you for catching this! We meant to say that every two elements have a *common* upper bound, and will revise this footnote in the next version.
>
> **Q2 (Connection to persistence modules).**
> Very interesting idea! Consistent sequences can indeed be viewed as functors from an appropriate combinatorial category to vector spaces. This connection is made precise in [2, Appendix A] and exploited throughout the representation stability literature, starting from the seminal work [3]. For example, the consistent sequences in our paper that are indexed by $\mathbb{N}$ with the usual order can be viewed as functors from FI, the category of finite sets with injective maps between them. Importantly, these categories are not simply the poset categories associated with $(\mathbb{N}, \preceq)$, since they contain multiple maps between any pair of objects. This multiplicity corresponds to the multiple embeddings between two vector spaces obtained by composing a fixed embedding $\varphi_{N,n}$ by all the group elements in $G_N$, and the combinatorics of these sets of embeddings has been exploited to study the relevant consistent sequences in [3]. Once again, it would be very interesting to identify and study categories corresponding to consistent sequences indexed by more general posets.
>
> [1] Levin et al. On Transferring Transferability. NeurIPS 2025.
>
> [2] Levin and Chandrasekaran. Dimension-Free Descriptions of Convex Sets. arXiv 2023.
>
> [3] Church et al. FI-modules and stability for representations of symmetric groups. Duke Math J. 2015.

---

> > ### Author Rebuttal · Reviewer_BxZS · 2026-03-31
> >
> > All of my concerns have been addressed.

---

> > > ### Author Response · Authors · 2026-04-07
> > >
> > > Thank you again for your thorough engagement with our paper!

---

### Official Review · Reviewer_VBA5 · 2026-03-09

**Soundness:** 3
**Presentation:** 4
**Significance:** 2
**Originality:** 3
**Overall Recommendation:** 3
**Confidence:** 3

**Summary:**

Based on a framework for any-dimensional networks, the paper provides a theory of universal approximation for invariant models with variable input sizes by embedding all finite-size inputs into an infinite-dimensional limit space and studying universality there.

**Compliance With Llm Reviewing Policy:**

Affirmed.

**Final Justification:**

I thank the authors for their response. While I understand that the primary focus of the paper is on the theoretical aspects, I am not fully convinced that the work can forgo experimental validation. In particular, I believe that some empirical results—at least simple or toy experiments—would help illustrate and support the theoretical claims.

Regarding the related work, four of [5–11] date back to earlier decades, while the remaining references are from 2019, 2021, and 2022. The models considered in those works are relatively standard and widely studied, whereas the models examined in this paper are more specialized and niche, and less widely understood. This further reinforces my view that some experimental evidence would be valuable to demonstrate the relevance of the theory in modern settings.

That said, I acknowledge that I may not be fully familiar with all aspects of this topic. I will therefore maintain my current score, but with a lower level of confidence, and will make sure to reflect this uncertainty during the discussion with the AC, particularly regarding the theory–empirical gap noted above.

**Key Questions For Authors:**

I have only one main concern, which is the lack of experimental validation, as mentioned in the Weaknesses section. Could the authors provide empirical results illustrating the practical advantages of enforcing the proposed continuity and universality properties for some of the architectures discussed in Section 4? For instance, it would be helpful to see experiments comparing the original architectures with the modified versions proposed in the paper.

Again, while I understand that the primary focus of the work is theoretical, the framework appears sufficiently concrete to allow at least some empirical observations. Without such evidence, the framework currently appears somewhat abstract and its practical implications remain unclear.

**Limitations:**

yes

**Strengths And Weaknesses:**

**Strengths.**

- Section 1 provides a clear and well-structured overview of the relevant literature and positions the work effectively within existing research.

- The paper is mathematically sophisticated and requires a solid functional-analysis background to fully appreciate the problem formulation. Nevertheless, the authors do a commendable job summarizing the necessary background in Section 2, largely building on prior work (especially [1]), which makes the framework reasonably accessible.

- The paper provides proofs for all stated results. While I did not verify the proofs in full detail, each main subsection (sets, graphs, and point clouds) includes proof sketches and follows a consistent analytical pipeline. The arguments, for me, appear mathematically sound and intuitively convincing.

**Weaknesses.** The paper does not include experimental results. While I understand that the primary contribution of the paper is theoretical, empirical validation could still be valuable in illustrating the practical implications of the theoretical findings. For example, in Section 4.1 (sets), the authors show that the original DeepSets architecture does not satisfy the desired continuity and universality properties under the proposed framework. They then introduce a modified variant of DeepSets that satisfies these properties. It would be useful to understand the practical consequences of this modification. In particular, it would be interesting to know:

- whether the proposed variant leads to measurable improvements in performance,

- whether there are additional computational costs (e.g., runtime or memory usage),

- or conversely, how the absence of these theoretical properties in the original architecture manifests itself in practical performance.

Such experiments would help clarify the practical relevance of the theoretical results.

---

[1] Eitan Levin, Yuxin Ma, Mateo Díaz, Soledad Villar. On Transferring Transferability: Towards a Theory for Size Generalization

---

> ### Author Rebuttal · Authors · 2026-03-30
>
> We sincerely thank the reviewer for their insightful comments.
>
> **W1/Q1 (Lack of experimental validation).**
> Thank you very much for your detailed suggestion! In brief, the architectures we consider in this paper are mostly small modifications of widely used and studied architectures from the literature. Their empirical performance is already well-demonstrated, and our paper instead studies their universality properties. Universality, in turn, is an important but fundamentally theoretical and asymptotic property, which is therefore intractable to demonstrate or test numerically. We proceed to elaborate on why this is the case.
>
> *Importance of universality.* All of our architectures are described by finitely many parameters, but are defined for infinitely many input sizes. We then ask whether such architectures are able to approximate arbitrary any-dimensional functions accepting inputs of any size. It is a priori unclear whether any architecture can achieve such an approximation, even if we let the number of its parameters grow. In fact, we identify in Section 4 several widely used any-dimensional architectures that fail to be universal. From this perspective, our architectures in Sec. 4 positively answer a fundamental theoretical question: Are there reasonable finitely-parametrized architectures that can approximate any continuous any-dimensional function?
>
> *Computational efficiency.* We do not claim that our methods are in any practical sense more competitive than existing alternatives. That being said, the models we propose are in most cases small modifications or compositions of existing models that have been widely tested in practice. The only exception is our graph architecture, which uses high-order tensors in the hidden layers. Our goal with this architecture is not to propose a computationally efficient or practical model; rather, our goal is to prove that universality is possible in this widely-studied setting. It is well-known that even in the finite-dimensional case, standard graph neural networks are not universal. In fact, they are at most as expressive as the WL-test [2]. Thus, it is natural to ask whether it is even possible to achieve universality using a reasonable, yet potentially expensive, model. We mention in passing that our architecture has significantly fewer parameters than those in [3, 4] ($2^{\binom{m}{2}}/m!$ versus $O(m^3)$; see response to Reviewer wBnb).
>
> *Testing our theory numerically.* Our theory concerns two important properties of any-dimensional models: continuity and universality. Testing continuity numerically is feasible, and indeed [1] shows, via extensive experiments, that imposing continuity yields transferability (performance of a model trained in low dimensions transfers to high dimensions), whereas discontinuous models fail to generalize across dimensions. We do not repeat these experiments here since transferability is not the contribution of our paper. Testing universality, on the other hand, is inherently intractable due to its asymptotic nature: universality asserts that for *every* continuous function on *every* compact set, an arbitrarily accurate approximation exists within the model class. No finite number of experiments can verify such a statement. For this reason, the classical universality results in deep learning [5, 6, 7], more recent universality results for invariant architectures [8, 9], and even the authoritative survey papers on the subject [10, 11], are all purely theoretical and contain no numerical experiments. Our paper fits squarely in this tradition.
>
> [1] Levin et al. On Transferring Transferability. NeurIPS 2025.
>
> [2] Xu et al. How Powerful are Graph Neural Networks? ICLR 2019.
>
> [3] NT and Maehara. Graph Homomorphism Convolution. ICML 2020.
>
> [4] Maehara and NT. A Simple Proof of the Universality of Invariant/Equivariant GNNs. arXiv:1910.03802.
>
> [5] Cybenko. Approximation by superpositions of a sigmoidal function. 1989.
>
> [6] Hornik et al. Multilayer feedforward networks are universal approximators. 1989.
>
> [7] Leshno et al. Multilayer feedforward networks with a nonpolynomial activation function can approximate any function. 1993.
>
> [8] Maron et al. On the universality of invariant networks. ICML 2019.
>
> [9] Yarotsky. Universal approximations of invariant maps by neural networks. 2022.
>
> [10] Pinkus. Approximation theory of the MLP model in neural networks. Acta Numerica 1999.
>
> [11] DeVore et al. Neural network approximation. Acta Numerica 2021.

---

> > ### Author Rebuttal · Reviewer_VBA5 · 2026-04-03
> >
> > I thank the authors for their response. While I understand that the primary focus of the paper is on the theoretical aspects, I am not fully convinced that the work can forgo experimental validation. In particular, I believe that some empirical results—at least simple or toy experiments—would help illustrate and support the theoretical claims.
> >
> > Regarding the related work, four of [5–11] date back to earlier decades, while the remaining references are from 2019, 2021, and 2022. The models considered in those works are relatively standard and widely studied, whereas the models examined in this paper are more specialized and niche, and less widely understood. This further reinforces my view that some experimental evidence would be valuable to demonstrate the relevance of the theory in modern settings.
> >
> > That said, I acknowledge that I may not be fully familiar with all aspects of this topic. I will therefore maintain my current score, but with a lower level of confidence, and will make sure to reflect this uncertainty during the discussion with the AC, particularly regarding the theory–empirical gap noted above.

---

> > > ### Author Response · Authors · 2026-04-07
> > >
> > > Thank you very much for your careful follow-up and for reconsidering your scores. We cited [5–11] in our previous rebuttal because they are authoritative references, but we agree that it is also useful to point to more recent universality papers. In fact, there are several modern works on universality that are primarily theoretical and do not rely on numerical experiments; see, for instance, [6,7,8,9]. At the same time, we stress that the architectures studied in our paper are, for the most part, small modifications of existing and well-tested models from the last several years. Specifically, the architecture in Sec. 4.1.1 is a slight modification of Deep Sets [1], the architecture in Sec. 4.1.2 is precisely the set-pooling architecture studied in [2], the architecture in Sec. 4.2 is closely related to higher-order invariant/equivariant graph architectures [3,4], and the architecture in Sec. 4.3 is a composition of the the equivariant scalar constructions in [5] and the IWN architecture in [7]. We agree that an empirical comparison of these architectures against existing alternatives would be useful for practitioners, though this lies outside the main scope of the present work.
> > >
> > > [1] Zaheer, Kottur, Ravanbakhsh, Póczos, Salakhutdinov, and Smola. Deep Sets. NeurIPS 2017.
> > >
> > > [2] Bueno and Hylton. On the Representation Power of Set Pooling Networks. NeurIPS 2021.
> > >
> > > [3] NT and Maehara. Graph Homomorphism Convolution. ICML 2020.
> > >
> > > [4] Maehara and NT. A Simple Proof of the Universality of Invariant/Equivariant GNNs. arXiv 2019.
> > >
> > > [5] Villar, Hogg, Storey-Fisher, Yao, and Blum-Smith. Scalars are Universal: Equivariant Machine Learning, Structured Like Classical Physics. NeurIPS 2021.
> > >
> > > [6] Pacini, Petrache, Lepri, Trivedi, and Walters. On Universality of Deep Equivariant Networks. ICLR 2026.
> > >
> > > [7] Herbst and Jegelka. Higher-Order Graphon Neural Networks: Approximation and Cut Distance. ICLR 2025.
> > >
> > > [8] Sonoda, Hashimoto, Ishikawa, and Ikeda. Deep Ridgelet Transform and Unified Universality Theorem for Deep and Shallow Joint-Group-Equivariant Machines. ICML 2025.
> > >
> > > [9] Li, Lin, and Shen. On the Universal Approximation Property of Deep Fully Convolutional Neural Networks. SIAM Journal on Mathematical Analysis, 2025.

---

### Official Review · Reviewer_uocf · 2026-03-10

**Soundness:** 3
**Presentation:** 3
**Significance:** 2
**Originality:** 3
**Overall Recommendation:** 3
**Confidence:** 3

**Summary:**

The authors study universality of any-dimensional (variable-size) invariant architectures. They start off by formalizing so-called compatible sequences of invariant maps, which are identified with a unique extension $f_\infty$ on a completed limit space $V_\infty$. Then we pass to the orbit space $V_\infty/G_\infty$ which is equipped with a symmetric metric d (see defs. 2.5-2.7 and eq. (1)). Afterwards, they define any-dimensional universality via standard uniform approximation on compact subsets of $V_\infty / G\_\infty$ (see eq. (2)), which they use with a Stone-Weierstrass-based recipe (in thm 3.1 plus the "step 1-3" part immediately after). The authors use this framework in the following four settings:

(i) Sets that have zero padding / sequence limits of $\ell^p$ (see sec. 4.1.1 on modified DeepSets, esp. eq. (5) with continuity in thm. 4.2 and universality in thm. 4.3).

(ii) Sets under duplication / Wasserstein $\mathcal{P}\_p$ limits (see sec. 4.1.2 on normalized DeepSets in eq. (6), continuity in thm. 4.5 and universality in thm. 4.6).

(iii) Graphs by using graphons in the cut metric $\delta_\square$ (in sec. 4.2 on graphon compactness (prop 4.7), and esp. function class $F_W$ in equations (9)-(10), and also continuity in thm. 4.9 and universality in thm. 4.10).

(iv) Point clouds by using the Gram to graphon map $W_X$ (from eq. (12) plus the IGNs from eq. (13)), which results in continuity/universality on $K_R/G_\infty$ (see thms. 4.12-4.13).

**Compliance With Llm Reviewing Policy:**

Affirmed.

**Final Justification:**

I find the paper sound and well-presented. However, I believe it has severe limitations in terms of applicability in practical scenarios, that made me question its significance. These issues were not completely addressed by the authors in the rebuttal. Thus, I still believe that the paper is not ready for publication, although I am not strongly convinced so.

**Key Questions For Authors:**

1. Users of this framework need to choose an embedding system and a norm. I think it would help a lot to see a concrete example where two natural-seeming choices give different continuity constraints, so the reader can see the importance of this.

2. Is the $\|X_{i:}\|^p$ weighting in Eq. (5) just a convenient way to sum or is there something deeper happening? Do you need tail decay of roughly something of that form?

4. It seems difficult to get translation invariance included in the point cloud setup. Does the compactness argument survive if it's included?

**Limitations:**

Yes

**Strengths And Weaknesses:**

## Strengths

The paper feels mathematically grounded. The chosen topology determines feasible architectures, which is one of the paper's main strengths. This is not just a remark but a tool for computation. (cf. Sec. 3, "Induced topology and activation functions", around lines 199-256)

The authors illustrate this quite well with their two limit sets, with $\ell^p$ and Wasserstein giving different compactness conditions, which then requires different pooling modifications.

For me, the most interesting part of the paper is the graphon story. In this case one commits to $\delta_\square$ and immediately pointwise nonlinearities are handled (linear is the exception). Interestingly, the homomorphism density algebra survives this process, which I found satisfying. The proof pattern itself (compacta, continuity, Stone-Weierstrass) is modular enough that I could imagine it being reused. It's also worth noting that the paper looks at various existing architectures, finds where they break, and suggests repairs. This is arguably more useful than the universality theorems themselves.

## Weaknesses

The graph universality is an asymptotic statement, and I'm not confident much of it survives in practice. The motif size (= tensor order) needs to grow to approximate arbitrary $\delta_\square$-continuous targets. That's fine in theory, but in practice an unbounded tensor order is likely of limited use. It would be interesting to know what you can get at order $k$ for some fixed $k$, which the paper doesn't touch. The integrals defining homomorphism densities also need to be estimated from a finite graph, and there's no discussion of what happens to the approximation error.

In my view, the paper also has a scope issue. In particular, $\delta_\square$ is a dense-graph topology. Many things people care about in graph ML (sparse graphs, anything depending on local neighborhoods) will not be continuous in $\delta_\square$. The paper doesn't really acknowledge this limitation head-on, and I think it should.

## Nitpicks

1. I was confused by the cut-norm normalization. The main text defines $|A|_\square$ with a $1/n^2$ factor (Sec. 4.2), but Appendix D.1 drops it. I believe they need to agree (Step-graphon identification).

2. Theorem 4.6 has an "asymptotically polynomial at $\pm\infty$" condition. This is supposed to hold on both tails, but sigmoid goes flat on one side, which seems like a problem. Given the ratio-to-polynomial definition in footnote 5, it seems not to cover Sigmoid (and also ReLU/Softplus at $-\infty$) as written.

3. The point cloud setup has permutation + $O(k)$ invariance but no translation invariance. This matters for most applications I can think of, and I believe it should be mentioned.

---

> ### Author Rebuttal · Authors · 2026-03-30
>
> We sincerely thank the reviewer for many helpful comments.
>
> **W1 (Universality is asymptotic / tensor order / integrals).**
> This is indeed an important concern! Universality is a fundamentally asymptotic property, not just in the graph case: classical results [1,2] require arbitrarily large width or depth, and no bounded-size architecture can be universal [3, Thm. 4.2]. In our any-dimensional setting, it is not clear a priori that finitely-parametrized architectures can approximate general any-dimensional functions to arbitrary precision, even as the number of parameters grows. In fact, we identify in Sec. 4 several widely used architectures that fail to be universal. Understanding those that do achieve universality is a key first step towards nonasymptotic guarantees, which we agree is a natural next step.
>
> *Fixed tensor order k.* With a fixed tensor order $k$, our architecture distinguishes graphs with differing $k$-node homomorphism densities, whereas GNNs and IGNs of order $k$ are tied to the $k$-WL test [4]. The relationship between $k$-WL and homomorphism densities is complex [5,6], and formally comparing the expressivity of these approaches at fixed tensor order is an interesting direction.
>
> *Approximating integrals.* Thank you for raising this question! We do not discretize the graphon integrals, as they naturally reduce to finite averages for finite graphs. A finite graph on $n$ nodes defines a step graphon, as we explain in Sec. D.1. When such a step graphon is plugged into the integral expressions in our paper, they reduce to finite averages over the entries of the adjecency matrix. Thus, the graphon (integral) formulation is the limiting object that we use to reason about universality across dimensions, but it does not introduce additional approximation error when the model is evaluated on a finite graph. We will clarify this point in the revision.
>
> **W2 (Scope: dense-graphs).**
> Thank you for pointing this out! We completely agree that our graph results are limited to dense graphs, and will acknowledge this in the revision. We focused on dense graph limits as their theory is quite mature at this point. We hope that our general recipe for proving universality---embedding into a limit space, identifying compact sets via symmetries, and applying Stone--Weierstrass---can be applied to some of the sparse graph limits in the literature (e.g., graphings, graphexes, graphops), and will suggest this as a direction for future work in the revision.
>
> **W3 (Cut-norm normalization).**
> We thank the reviewer for catching this typo. The factor $1/n^2$ in Section 4.2 is necessary to ensure that the graph norm is compatible with duplication embeddings. We have corrected Appendix D.1.
>
> **W4 (Sigmoid).**
> Thank you for this question! Sigmoids are indeed asymptotically polynomial---they satisfy the condition in footnote 5 with the constant polynomials $P_{\pm}(t) = \pm 1$. We will explicitly point this out.
>
> **W5/Q3 (Translation invariance).**
> Thank you for raising this issue. We can seamlessly handle translations by centering the point cloud before applying our architecture, which accounts for both symmetries. Our compactness argument goes through verbatim. Sec 4.3 will be updated with a discussion about it.
>
> **Q1 (Different embeddings/norms give different topology).**
> We completely agree that such an example would be very instructive! Our submission already contains two such examples, which we will highlight more explicitly. First, for sets, the different embeddings and norms in Sections 4.1.1 and 4.1.2 act on the same spaces $V_n = \mathbb{R}^{n \times k}$, but the choice of embedding determines whether architectures are defined on sequences or measures, and the norm determines which sequences or measures comprise the limit space. Second, the cut norm and $L^2$ norm on graphons (Sections 4.2 and 4.3) yield different continuity constraints: pointwise activations are continuous in $L^2$ but not in cut norm, necessitating an architecture such as Eq. 6 for universality in cut metric.
>
> **Q2 (Norm weighting in Eq. 5).**
> Thank you for asking, there is indeed something deeper happening here! The $|X_{i:}|^p$ weighting in Eq. (5) is necessary for continuity: without it, the standard DeepSets architecture in Eq. (4) is discontinuous as it can diverge for sequences in $\ell_p(\mathbb{R}^k)$.
>
> [1] Cybenko. Approximation by superpositions of a sigmoidal function. 1989.
>
> [2] Hanin and Sellke. Approximating Continuous Functions by ReLU Nets of Any Width. MSML 2019.
>
> [3] DeVore et al. Optimal nonlinear approximation. Manuscripta Math 1989.
>
> [4] Maron et al. Provably Powerful Graph Networks. NeurIPS 2019.
>
> [5] Chen et al. Can Graph Neural Networks Count Substructures? NeurIPS 2020.
>
> [6] Lanzinger and Barcelo. On the Power of the Weisfeiler-Leman Test for Graph Motif Parameters. ICLR 2024.

---

> > ### Author Rebuttal · Reviewer_uocf · 2026-04-01
> >
> > Thank you for the detailed rebuttal. The clarification about how the graphon expressions evaluate on finite graphs, and the correction to the cut-norm normalization, are helpful.
> >
> > That said, my main concerns remain unchanged. The graph contribution is still restricted to the dense-graph / cut-metric setting, and the practically relevant fixed tensor-order regime is still left open. So while I appreciate the clarifications and planned revisions, I would like to keep my overall score unchanged.

---

> > > ### Author Response · Authors · 2026-04-07
> > >
> > > Thank you again for your careful reading and for clarifying your position. We fully agree that extending these results beyond the dense-graph / cut-metric setting, as well as understanding the expressivity of bounded tensor-order architectures, are both important and practically relevant directions for future work. At the same time, we would respectfully emphasize that the goal of our paper is to develop a systematic framework for studying any-dimensional universality that is applicable across a variety of application domains. To the best of our knowledge, such a framework has not previously been developed in the literature. Our approach is intentionally built by leveraging and unifying existing limit-space viewpoints for sets, graphs, and point clouds, rather than by resolving all limitations of the corresponding domain-specific theories.
> > >
> > > In the graph setting, we focus on the graphon / cut-metric framework because it is a well-established and influential limit theory that has already served as a useful foundation for the analysis of graph neural networks and related approximation questions [1,2,3,4,5]. In particular, even universal approximation has recently been studied in this setting [5], which supports its value as a natural first testbed for any-dimensional graph learning.
> > >
> > > Likewise, we agree that the bounded tensor-order regime is important, and there are already meaningful results in this direction [5,6]. Our point is not that this regime is unimportant, but rather that asymptotic universality is a natural first step: before asking for non-asymptotic or fixed-order guarantees, one must first understand whether finitely parameterized architectures can approximate arbitrary any-dimensional functions at all. As we explain in this paper, and as raised already in [5], the answer to this question is involved even when arbitrary tensor orders are allowed. Our contribution is therefore complementary to, rather than in competition with, efforts aimed at sparse limits or bounded tensor-order expressivity.
> > >
> > > [1] Ruiz, Gama, Ribeiro, and Bruna. Graphon Neural Networks and the Transferability of Graph Neural Networks. NeurIPS 2020.
> > >
> > > [2] Maskey, Levie, Lee, and Kutyniok. Transferability of Graph Neural Networks: An Extended Graphon Approach. Applied and Computational Harmonic Analysis, 2023.
> > >
> > > [3] Levie. A Graphon-Signal Analysis of Graph Neural Networks. NeurIPS 2023.
> > >
> > > [4] Ruiz, Chamon, and Ribeiro. Transferability Properties of Graph Neural Networks. IEEE Transactions on Signal Processing, 2023.
> > >
> > > [5] Herbst and Jegelka. Higher-Order Graphon Neural Networks: Approximation and Cut Distance. ICLR 2025.
> > >
> > > [6] Lanzinger and Barceló. On the Power of the Weisfeiler-Leman Test for Graph Motif Parameters. ICLR 2024.

---

### Official Review · Reviewer_wBnb · 2026-03-12

**Soundness:** 3
**Presentation:** 3
**Significance:** 2
**Originality:** 3
**Overall Recommendation:** 5
**Confidence:** 5

**Summary:**

This paper studies universality for invariant machine learning models that operate on inputs of arbitrary size. While standard universal approximation only works on a fixed domain, any-dimensional architectures are sequences of functions defined on growing input spaces. To establish universal approximation in such cases, the authors embed all finite (normed) domains of an any-dimensional model into an infinite-dimensional limit space by identifying smaller instances with larger ones. The core contribution of the paper is a recipe for proving universality in this setting by first defining an appropriate topology of the limit space (which is crucial in the infinite-dimensional space), ensuring the model is continuous w.r.t. this topology, and then invoke standard Stone-Weierstrass arguments on compact subsets of the limit space. The authors invoke this framework for sets, graphs, and point clouds, and propose architectures that are continuous and universal w.r.t. a natural limit topology, each.

**Compliance With Llm Reviewing Policy:**

Affirmed.

**Final Justification:**

See also rebuttal acknowledgement. While I think that my point about the contribution being largely conceptual still stands, my point about the neural parameterization in the graph case has been adequately addressed.

**Key Questions For Authors:**

- In the graph case, how should one think about the practical value of the introduced parameterization, as the proof reduces to recovering simple graph homomorphism densities?
- Just a minor contextual comment, but the paper emphasizes that topology matters in the infinite-dimensional case. This is certainly true, but topology can matter in finite models as well for quotient/pseudometric structures and one needs point separation for Stone-Weierstrass arguments (which is especially common in the graph case, see, e.g., [4,5]). In this sense, the claim in lines 49/55 could maybe be substantiated a bit more.

[4] Ching-Yao Chuang, Stefanie Jegelka. Tree Mover's Distance: Bridging Graph Metrics and Stability of Graph Neural Networks. NeurIPS 2022.

[5] Levi Rauchwerger, Stefanie Jegelka, Ron Levie. Generalization, expressivity, and universality of graph neural networks on attributed graphs. ICLR 2025.

**Limitations:**

yes

**Strengths And Weaknesses:**

Strengths:

- The framing is clear: For size-varying invariant models, the right object is not a family of unrelated finite-dimensional approximators but a compatible sequence with a common extension to a limit space. The point that the topology induced by the chosen norm on the limit space matters materially is useful (though in some domains well known, e.g., from graph limit theory).
- Generally, the paper is well motivated and easy to follow for a theory audience. The main body is quite focused and keeps the exposition readable, as many of the technical details are deferred to the appendix.
- I found the construction for point clouds in section 4.3 particularly appealing. Among the three instantiations of the framework, mapping a point cloud to a “Gram graphon” and then invoking graphon neural networks felt the most original to me.

Weaknesses:

- The contribution appears more conceptual than practical, namely the results are mathematically interesting, but it is not made clear how relevant they are for realistic models of finite capacity. In the set case, the argument relies heavily on standard universal approximation with arbitrarily large width. In the graph case, the authors note that the tensor order, width, and depth grow with motif size. The results do not make any statement about approximation quality under realistic architectural constraints.
- The paper explicitly adopts the framework of [1], Section 2 largely follows it with the setup, and the paper spends substantial space introducing this existing framework before getting to the new universality results.
- In the graph section, the actual universal class is essentially constructed to recover linear combinations of simple graph homomorphism densities, and the neural parameterization that realizes it is acknowledged to be rather involved. It is less clear to me whether the parameterization would be preferable in practice to working in the homomorphism density basis itself, as, e.g., suggested in [2,3] with similar arguments.

[1] Eitan Levin, Yuxin Ma, Mateo Diaz Diaz, Soledad Villar. On Transferring Transferability: Towards a Theory for Size Generalization. NeurIPS 2025.

[2] Hoang NT, Takanori Maehara. Graph Homomorphism Convolution. ICML 2020.

[3] Takanori Maehara, Hoang NT. A Simple Proof of the Universality of Invariant/Equivariant Graph Neural Networks. https://arxiv.org/pdf/1910.03802

Miscellaneous:

- If I am not mistaken, the “closed” in the theorem statement of Stone-Weierstrass, line 231, should be removed. Maybe this originally referred to complex conjugation in a more general case, but here it risks being confused with topological closedness.

---

> ### Author Rebuttal · Authors · 2026-03-30
>
> We thank the reviewer for providing many insightful comments.
>
> **W1 (Conceptual vs. practical, unbounded capacity).**
> This is a very important concern!
> Even in the classical setting, with fixed input size, the size of the architecture must grow to approximate an arbitrary continuous function to arbitrary precision. Indeed, classical results [6] require arbitrarily large width while more modern ones [7] require arbitrarily large depth to achieve universality. Our any-dimensional universality results naturally also require the models to grow in size. This is not an artifact of the theory, as one can show that no bounded-size architecture can be universal, see [9]. Likewise, there are fundamental obstructions limiting the expressivity of standard GNNs [8], which necessitate the use of higher-order tensors in our graph architectures.
>
> Importantly, each of our models is described by finitely-many parameters. It is not clear a priori that such a model might be able to approximate any-dimensional functions, even if we let the number of parameters to grow. In fact, we identify in Sec. 4 several widely used models that fail to be universal.
> From this perspective, our models in Sec. 4 answer a fundamental question: Are there reasonable finitely-parametrized models that can approximate any continuous any-dimensional function?
> Understanding such architectures is a key first step to further analyzing their approximation capabilities in a non-asymptotic setting, and we agree that this is a natural and important next step for the field. We will clarify the above in the revision.
>
> **W2 (Background takes space).**
> Thank you for the helpful feedback! As we explain in Sec. 1, and will emphasize in the revision, there is a basic question we need to answer in order to study any-dimensional universality: How do we think about functions that take inputs of different sizes, and in what sense can such an architecture be universal? We use [1] to define the class of functions and architectures that we consider, so we deemed it essential to review the basic elements of that framework.
> To ensure readers reach our core contributions as quickly as possible, we carefully minimized the background material in the main text, relegating details to Appendix A.
>
> **W3/Q1 (Value of parameterization vs. homomorphism density basis).**
> Thank you for this insightful question. As you note, the proof of universality for our architecture reduces to recovering simple graph homomorphism densities. However, our architecture offers two key advantages over working in the homomorphism density basis directly.
>
> *Parameter efficiency.* Each model $h_k^{a,b}(\cdot)$ in Eq. (9) is defined by $k(k-1)$ parameters, and the span of all $k \leq m$ parameterizes the space of all homomorphism densities of simple graphs on $m$ nodes. Taking a linear combination of $h_1^{a,b}, \ldots, h_m^{a,b}$ requires $m$ additional coefficients, giving a total of $m + \sum_{k=1}^{m} k(k-1) = O(m^3)$ parameters to span the above space. In contrast, the basis approach in [2,3] requires explicitly enumerating all nonisomorphic simple graphs on $m$ nodes, whose count grows at least as $2^{\binom{m}{2}}/m!$ (see https://oeis.org/A000088). Our parameterization thus achieves an exponential compression thanks to the nonlinearity of Eq. (9).
>
> *Gradient-based training.* The basis approach also requires computing each homomorphism density separately for every motif, which is costly. Our parameterization avoids this by encoding graph patterns implicitly through the learnable coefficients $a_{ij}, b_{ij}$. Moreover, as detailed in Appendix D.4, our model decomposes into a neural-network-like deep architecture making it directly amenable to gradient-based optimization.
>
>
> **W4 (Stone-Weierstrass "closed").**
> We completely agree with the reviewer and thank them for catching this typo.
> We have removed it from the revised manuscript.
>
> **Q2 (Topology matters in finite dimensions too).**
> Thank you for this remark, we will certainly clarify our claim in the revision. The claim in lines 49/55 refers narrowly to the fact that all norms induce the same topology in finite dimensions.
>
> [1] Levin et al. On Transferring Transferability. NeurIPS 2025.
>
> [2] NT and Maehara. Graph Homomorphism Convolution. ICML 2020.
>
> [3] Maehara and NT. A Simple Proof of the Universality of Invariant/Equivariant Graph Neural Networks. arXiv:1910.03802.
>
> [4] Chuang and Jegelka. Tree Mover's Distance. NeurIPS 2022.
>
> [5] Rauchwerger et al. Generalization, expressivity, and universality of GNNs on attributed graphs. ICLR 2025.
>
> [6] Cybenko. Approximation by superpositions of a sigmoidal function. 1989.
>
> [7] Hanin and Sellke. Approximating Continuous Functions by ReLU Nets of Any Width. MSML 2019.
>
> [8] Xu et al. How Powerful are Graph Neural Networks? ICLR 2019.
>
> [9] DeVore et al. Optimal nonlinear approximation. Manuscripta Math 1989.

---

> > ### Author Rebuttal · Reviewer_wBnb · 2026-04-03
> >
> > I thank the authors for their thorough rebuttal. I particularly appreciate the response to W3/Q1, and think that including this discussion about parameter efficiency in a revision will make the contribution towards the graph case of the universality framework clearer. Thus, I vote for acceptance and will raise my score.

---

> > > ### Author Response · Authors · 2026-04-07
> > >
> > > Thank you very much for updating your score! We will be sure to include the discussion on parameter efficiency in the revision.

---

### Decision · Program_Chairs · 2026-04-30

**Decision:**

Accept (regular)

**Comment:**

The authors consider ML models defined on inputs of any size, called any-dimensional models. They study the universality of such models with a new idea: they identify sequences of dimension-indexed functions as a unique function in an infinite-dimensional space. They propose to use topology and symmetries in such a space, and show that while many existing models are not universal, they can propose modifications to achieve universality.

The reviewers found the paper interesting and well presented. The paper is theoretically relevant and of interest to the ML community. However, the message of the paper and its applicability to practical scenarios are still not well developed in the paper:

> Reviewer uocf: I believe it has severe limitations in terms of applicability in practical scenarios, that made me question its significance. These issues were not completely addressed by the authors in the rebuttal.

> Reviewer wBnb: While I think that my point about the contribution being largely conceptual still stands, my point about the neural parameterization in the graph case has been adequately addressed.


> Reviewer VBA5: While I understand that the primary focus of the paper is on the theoretical aspects, I am not fully convinced that the work can forgo experimental validation. In particular, I believe that some empirical results—at least simple or toy experiments—would help illustrate and support the theoretical claims.


> Reviewer BxZS: I also saw that other reviewers raised concerns about practical implementation of the methods.


The reviewers all pointed out unanimously that having experiments, even small-scale ones to show the relevance of the results to practice, is essential. Other than that, I agree with the reviewers that the theoretical findings are very interesting, and I strongly suggest the authors to take care of the reviewers' concerns.

One additional concern is that the authors claim certain models are not universal and propose a method to address this issue. However, the paper does not provide a sufficiently clear or practical message for a broader ML audience on which types of functions actually break universality. In particular, it would be helpful to include more explicit examples with intuitive explanations of the properties that lead to this limitation, for readers with an empirical viewpoint. While these aspects are developed theoretically in the paper, and I appreciate it, presenting them in a more accessible manner can significantly imrpve both the clarity and the overall impact of the work.